# Deep generative modeling of transcriptional dynamics for RNA velocity analysis in single cells

Adam Gayoso [1,10], Philipp Weiler [2,3,10], Mohammad Lotfollahi [2,4], Dominik Klein[2,3], Justin Hong [1,9], Aaron Streets [1,5,6], Fabian J. Theis [2,3,7] ✉ & Nir Yosef [1,8] ✉

RNA velocity has been rapidly adopted to guide interpretation of transcriptional dynamics in snapshot single-cell data; however, current approaches for estimating RNA velocity lack effective strategies for quantifying uncertainty and determining the overall applicability to the system of interest. Here, we present veloVI (velocity variational inference), a deep generative modeling framework for estimating RNA velocity. veloVI learns a gene-specific dynamical model of RNA metabolism and provides a transcriptome-wide quantification of velocity uncertainty. We show that veloVI compares favorably to previous approaches with respect to goodness of fit, consistency across transcriptionally similar cells and stability across preprocessing pipelines for quantifying RNA abundance. Further, we demonstrate that veloVI's posterior velocity uncertainty can be used to assess whether velocity analysis is appropriate for a given dataset. Finally, we highlight veloVI as a flexible framework for modeling transcriptional dynamics by adapting the underlying dynamical model to use time-dependent transcription rates.

Advances in single-cell RNA sequencing (scRNA-seq) technologies have facilitated the high-resolution dissection of the mechanisms underlying cellular differentiation and other temporal processes[1–3]. Although scRNA-seq is a destructive assay, a widely used set of computational approaches leverage the asynchronous nature of dynamical biological processes to order cells along a so-called pseudotime in the task of trajectory inference[4–7]. Traditional methods for trajectory inference typically require the initial state of the underlying biological process to be known and use manifold learning to determine a metric space in which distances capture changes in differentiation state.

Recently, RNA velocity has emerged as a bottom-up, mechanistic approach for the trajectory inference task. RNA velocity, which describes the change of spliced messenger RNA (mRNA) over time, makes use of concomitant detection of unspliced and spliced RNA transcripts with standard scRNA-seq protocols[8]. Upon estimation, RNA velocity is typically incorporated into analyses in two ways: (1) inferring a cell-specific differentiation pseudotime or (2) constructing a transition matrix inducing a Markov chain over the data to determine initial, transient and terminal subpopulations of cells[9].

There are currently two popular methods for estimating RNA velocity. The first, referred to as the steady-state model, assumes (1)

[1]Center for Computational Biology, University of California, Berkeley, Berkeley, CA, USA. [2]Institute of Computational Biology, Helmholtz Center Munich, Munich, Germany. [3]Department of Mathematics, Technical University of Munich, Munich, Germany. [4]Wellcome Sanger Institute, Cambridge, UK. [5]Department of Bioengineering, University of California, Berkeley, Berkeley, CA, USA. [6]Chan Zuckerberg Biohub, San Francisco, CA, USA. [7]TUM School of Life Sciences Weihenstephan, Technical University of Munich, Munich, Germany. [8]Department of Electrical Engineering and Computer Sciences, University of California, Berkeley, Berkeley, CA, USA. [9]Present address: Department of Computer Science, Columbia University, New York, NY, USA. [10]These authors contributed equally: Adam Gayoso, Philipp Weiler. ✉e-mail: fabian.theis@helmholtz-munich.de; niryosef@berkeley.edu

constant rates of transcription and degradation of RNA; (2) a single, global splicing rate[8,10]; (3) that the cellular dynamics reached an equilibrium in the induction phase and do not include basal transcription; and (4) gene-wise independence. The second method, referred to as the EM model, was previously described and implemented in the scVelo package[11]. The EM model relaxes the assumption of the system having reached a steady-state, infers the full set of transcriptional parameters and estimates a latent time per cell, per gene by formulating the problem in an expectation-maximization (EM) framework.

While these approaches for estimating RNA velocity have been successfully used to interpret single-cell dynamics[12,13], they also suffer from limitations derived from their modeling assumptions and downstream usage[14–17]. For example, both methods lack a global notion of uncertainty. Thus, assessing the robustness of the RNA velocity estimate, or deciding to what extent velocity analysis is appropriate for a given dataset can be difficult. Although the EM model can be used to rank putative 'driving' genes by their likelihood, there is no direct connection between gene likelihood, visualization and correctness. For example, in the case of dentate gyrus neurogenesis, visualization of RNA velocity suggests that granule mature cells develop into their immature counterparts even though a selection of high likelihood genes suggests the reverse (correct) dynamics[11].

Estimation of RNA velocity with current approaches is also tightly coupled to the parameterization of the differential equations underlying transcription. Assumptions such as constant transcription, splicing and degradation rates may be too simple to explain dynamics that arise in multi-lineage[14] or even single-lineage[18] cell differentiation. The methods outlined to estimate RNA velocity lack extensibility and flexibility to adapt to more complicated, real-world scenarios. Emerging technologies such as VASA-seq[19], which have greater sensitivity for unspliced RNA detection, may provide sufficient signal to fit more complex models.

To address these issues, we present veloVI (velocity variational inference), a deep generative model for estimating RNA velocity. VeloVI reformulates the inference of RNA velocity via a model that shares information between all cells and genes, while learning the same quantities, namely kinetic parameters and latent time, as in the EM model. This reformulation leverages advances in deep generative modeling[20], which have become integral to many single-cell omics analytical tasks such as multimodal data integration[21,22], perturbation modeling[23,24] and data correction[25]. As its output, veloVI returns an empirical posterior distribution of RNA velocity (matrix of cells by genes by posterior samples), which can be incorporated into the downstream analysis of the results. Here, we show that veloVI represents a substantial improvement over the EM model in terms of fit to the data. Additionally, it provides a layer of interpretation and model criticism lacking from previous methods while also greatly improving flexibility for model extensions.

We use veloVI to enhance analyses of velocity at the level of cells, genes and whole datasets. At the level of a cell, veloVI illuminates cell states that have directionality estimated with high uncertainty, which adds a notion of confidence to the velocity stream and highlights regions of the phenotypic manifold that warrant further investigation and more careful interpretation. We couple this analysis with a metric called velocity coherence that explains the extent to which a gene agrees/disagrees with the inferred directionality. At the level of genes and datasets, we propose a permutation-based technique using veloVI that can identify partially observed dynamics or systems in steady states. This can be used to determine the extent to which RNA velocity analysis is suitable for a particular dataset.

Finally, veloVI is an extensible framework to fit more sophisticated transcriptional models. We highlight this flexibility by extending the current transcriptional model with a time-dependent transcription rate and show how this extension can improve the model fit.

## Results

### Variational inference for estimating RNA velocity

VeloVI posits that the unspliced and spliced abundances of RNA for each gene in a cell are generated as a function of kinetic parameters (transcription, splicing and degradation rates), a latent time and a latent transcriptional state (induction state, repression state and their respective steady states). Additionally, veloVI posits that each gene's latent times (per cell) are tied via a low-dimensional latent variable that we call the cell representation. These representations capture the notion that the observed state of a cell is a composition of multiple concomitant processes that together span the phenotypic manifold[1]. This modeling choice is justified by the observation that with the EM model, which is fit independently per gene, the inferred latent time matrix (of shape cells by genes) has a low-rank structure (but notably, not rank one; Extended Data Fig. 1).

The complete architecture of veloVI manifests as a variational autoencoder[26]. The encoder neural networks take the unspliced and spliced abundances of a cell as input and output the posterior parameters for the cell representation and latent transcriptional state variables. The gene-wise, state-specific, latent time is parametrized by a neural network that takes a sample of the cell representation as input. The likelihood of cellular unspliced and spliced abundances is then a function of the latent time, the kinetic rate parameters and the state assignment probabilities (Fig. 1a and Methods). The model's parameters are optimized simultaneously using standard gradient-based procedures. After optimization, the cell-gene-specific velocity is computed as a function of the degradation rate, the splicing rate and the fitted unspliced and spliced abundances, which directly incorporate the posterior distributions over time and transcriptional state.

As a Bayesian deep generative model, veloVI can output a posterior distribution over velocities at the cell-gene level. This distribution can be used to quantify an intrinsic uncertainty over the first-order directions that a cell can take in the gene space. In downstream analyses, velocity is often used to construct a cell–cell transition matrix that reweights the edges of a nearest-neighbors graph according to the similarity of the first-order displacement of a cell and its neighborhood[8,11]. By piping posterior velocity samples through this computation, we also quantify an extrinsic uncertainty, which reflects both the intrinsic uncertainty and the variability among the cell's neighbors in gene space (Fig. 1b and Methods). In contrast, the EM model and steady-state model do not carry any explicit notion of uncertainty. Indeed, both previous models only allow evaluating an uncertainty post-hoc based on quantifying velocity variation over a cell's neighbors[9]. Finally, a point estimate of the velocity averaged over samples for a cell allows veloVI's output to be used directly in scVelo's downstream visualization and graph construction functionalities as well as other packages building upon scVelo[9,27].

### veloVI improves data fit over the EM model and is stable

We performed a multifaceted analysis to evaluate veloVI's ability to robustly fit transcriptional dynamics across a range of simulated and real datasets, comparing to both the EM model and the steady-state formulation of RNA velocity as implemented in the scVelo package[11].

We first assessed each model's ability to recover kinetic parameters in simulated data (Methods). With an increasing number of observations, veloVI outperformed the EM model and was better than the steady-state model in recovering the simulated ratio of degradation and splicing rate for each gene (Supplementary Fig. 1a). Similarly, veloVI's inferred latent time and velocity correlated significantly better (two-sided Welch's *t*-test, *P* < 0.001) with ground truth compared to EM estimates when simulating data with parameters previously estimated on real data (Methods and Fig. 2a). It is notable that these simulations reflect an idealized scenario as cells are simulated via the EM model generative process, which assumes gene-wise independence, induction followed by repression states and a single lineage

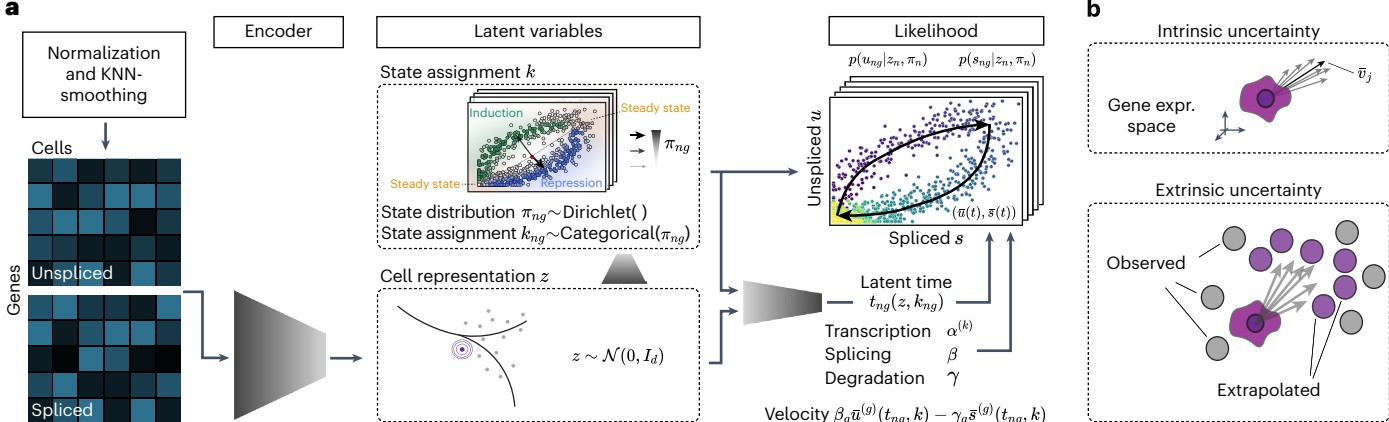

**Fig. 1 | Overview of the veloVI model. a**, veloVI encodes the unspliced and spliced abundances into the cell representation through a neural network. This cell representation is used to further encode the transcriptional state assignment of each cell/gene. Using the cell representation as input, the decoder neural network outputs a cell-gene-state specific latent time. The likelihood is a function of the latent time, kinetic rates (rates of transcription ($\alpha$), splicing ($\beta$) and degradation ($\gamma$)) and uncertainty over the state assignment. The model is optimized end-to-end using stochastic variational inference techniques. **b**, The variational inference formulation quantifies the intrinsic uncertainty of a velocity estimate by sampling from the posterior distribution and measuring variability around the mean velocity vector. This notion is contrasted by its extrinsic counterpart, which quantifies variation of velocity extrapolated cell states within a cell's neighborhood defined by transcriptomic similarity. KNN, $k$-nearest neighbor.

(Methods). Nonetheless, veloVI outperforms the EM model even in these EM-favorable conditions. We also benchmarked the runtime of veloVI and EM model. For this comparison, we ran both models on sub-samples of a mouse retina dataset[28] containing approximately 114,000 cells. Across multiple subsamples, inference was substantially faster using veloVI compared to the EM model (Supplementary Fig. 1b). Specifically, considering 20,000 cells, veloVI achieved a fivefold speed-up.

To further validate the accuracy of veloVI, we compared veloVI and the EM model on cell-cycle datasets of fluorescent ubiquitination-based cell-cycle indicator (FUCCI) RPE1 and U2OS cells[13,29] as it offers orthogonal validation of directionality/time via a protein-derived cell-cycle score (Fig. 2b). To assess model performance, we first compared the local consistency of the velocity vector fields generated by each model. This consistency measure quantifies the extent to which the velocities of cells with similar transcriptomic profiles (nearest neighbors) agree and relies on the assumption that velocities change smoothly over the phenotypic manifold. Compared to the EM model, veloVI achieves a higher velocity consistency (Fig. 2c). We also tested whether the direction of the velocity at the gene level aligns with a ground truth heuristic based on the cell cycle (Methods). As before, veloVI yielded consistent results and outperformed the EM model (RPE1 (resp. U2OS), 66% (resp. 68%) genes have higher velocity sign accuracy under veloVI; Fig. 2d) significantly (one-sided Welch's $t$-test, $P < 0.001$). As a complementary validation of these findings, we confirmed that the velocities of individual genes inferred by veloVI change more smoothly (are less noisy) with respect to the ground truth 'time' compared to the EM model (RPE1 (resp. U2OS), 78% (resp. 65%) genes have higher $R^2$ under veloVI) (Fig. 2e, Supplementary Fig. 2 and Methods).

We then evaluated the stability of velocity estimates on real datasets processed with 12 different RNA abundance quantification algorithms[8,28,30-33], based on previous work that highlighted general inconsistencies in velocity estimation[34] (Methods). To do so, we measured the correlation of velocity of each cell between pairs of quantification flavors on five benchmarking examples, namely pancreas endocrinogenesis at embryonic day 15.5 (ref. 35) as well as datasets of spermatogenesis[36], mouse developing dentate gyrus[37], the prefrontal cortex of a mouse[38] and 21–22-month-old mouse brains[39]. When aggregating these correlations for each pair of quantification algorithms, veloVI scored both a higher mean correlation and lower variance compared to the EM model. Compared to the much simpler steady-state

model, veloVI tended to have a similar mean correlation, but with lower variance (Fig. 3a, Extended Data Fig. 2 and Supplementary Figs. 3–7).

To assess how well the inferred dynamics reflect the observed data, we computed the mean squared error (MSE) of the fit for the unspliced and spliced abundances and compared the MSE to that of the EM model on a selection of datasets (Supplementary Table 1). For each dataset, we computed the ratio of the MSE for veloVI and the EM model at the level of a gene. VeloVI had better performance for a majority of the genes in each dataset (Fig. 3b). Additionally, across all datasets, veloVI had higher velocity consistency among cells (Fig. 3b). We attribute this increase to the explicit low-dimensional modeling in veloVI that shares statistical strength across all cells and genes.

Despite sharing many model assumptions, the velocities estimated for a gene with veloVI were partially correlated on average with their EM counterpart (Fig. 3b). To highlight the differences in velocity estimation at the level of individual genes, we examined *Sulf2*, a marker of endocrine progenitor cells and *Top2a*, a cell-cycle marker, in the pancreas dataset (Fig. 3c). For both of these genes, the EM model predicted a wide range of velocities for cells that had near-zero unspliced and spliced abundances. For example, terminal beta cells had substantially positive velocity under the EM model for *Sulf2* despite being located at the bottom-left of the phase portrait (defined as the scatter-plot of unspliced versus spliced abundance of a gene) and with known development occurring later than endocrine progenitors and pre-endocrine cells. In the case of veloVI, beta cells had nearly zero velocity, reflecting their belonging to the putative repression steady state for this gene. We attribute this result to veloVI's velocity directly marginalizing over the latent cell representations, which explicitly incorporates the probability that a cell belongs to induction, repression, or their respective steady states (Methods). We observed similar results for *Top2a*, in which cell types without a strong cell-cycle signature and near-zero unspliced/spliced abundance had positive velocity in the EM model, but near-zero velocity using veloVI.

## veloVI enables interpretable velocity analysis
We then investigated how the uncertainty in the velocity estimates of veloVI could be used to scrutinize its output, both at the level of cells (which might be incorrectly modeled) and at the level of individual genes (which might be inconsistent with the aggregated, cell-level output). We used this uncertainty to (1) measure the variability in the

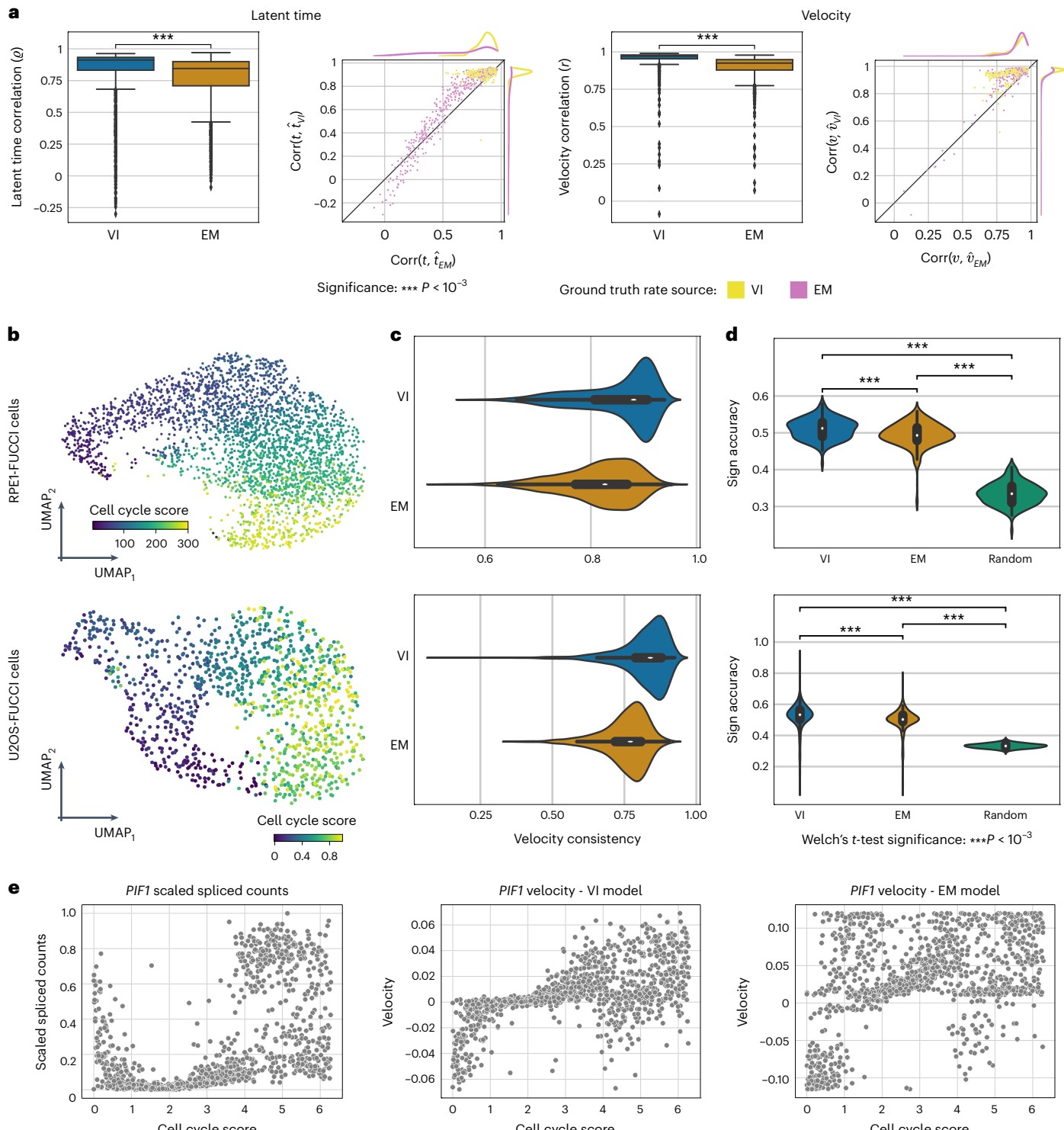

**Fig. 2 | Benchmarking of velocity and latent time recovery. a**, Accuracy of VI and EM models on recovering ground truth latent time and velocity. For each quantity, we provide results in the form of box plots ($n = 1,074$ genes) and scatter plots (VI model $y$ axis, EM model $x$ axis). Rate parameters were estimated by each model on the pancreas dataset. Following this, a subset of these parameters was used to simulate new data, which each model was fitted to. In these plots, each point corresponds to a gene, where the value is the correlation between the estimated time/velocity versus the ground truth. The color coding in the scatter plots indicates whether the simulated rate parameters were derived from the original EM or VI model fit. Box plots indicate the median (center line), interquartile range (hinges) and whiskers at 1.5× interquartile range. **b**, UMAP colored with FUCCI-derived cell-cycle score for the RPE1- (top) and U2OS-FUCCI cells (bottom). **c**, Comparison of the velocity consistency from veloVI (blue) and the EM model (orange). The cell-wise velocity consistency in case of RPE1-FUCCI cells is shown on top, their U2OS counterparts on the bottom ($n = 2,793$ cells, top; $n = 1,146$ cells, bottom). Box plots indicate the median (center line) and interquartile range (hinges). **d**, Comparison of veloVI's and the EM model's estimated velocities. The violin plots show the log-transformed ratio of each method's velocity sign accuracy, which were computed per gene. Box plots indicate the median (center line) and interquartile range (hinges) ($n = 140$ genes, top; $n = 395$ genes, bottom). Significance was assessed with a one-sided Welch's $t$-test ($P < 0.001$). **e**, Spliced abundance of *PIF1* versus cell cycle score for each cell in the U2OS-FUCCI dataset (left). Estimated velocity in *PIF1* using the VI (middle) and EM model (right) plotted against the cell-cycle score.

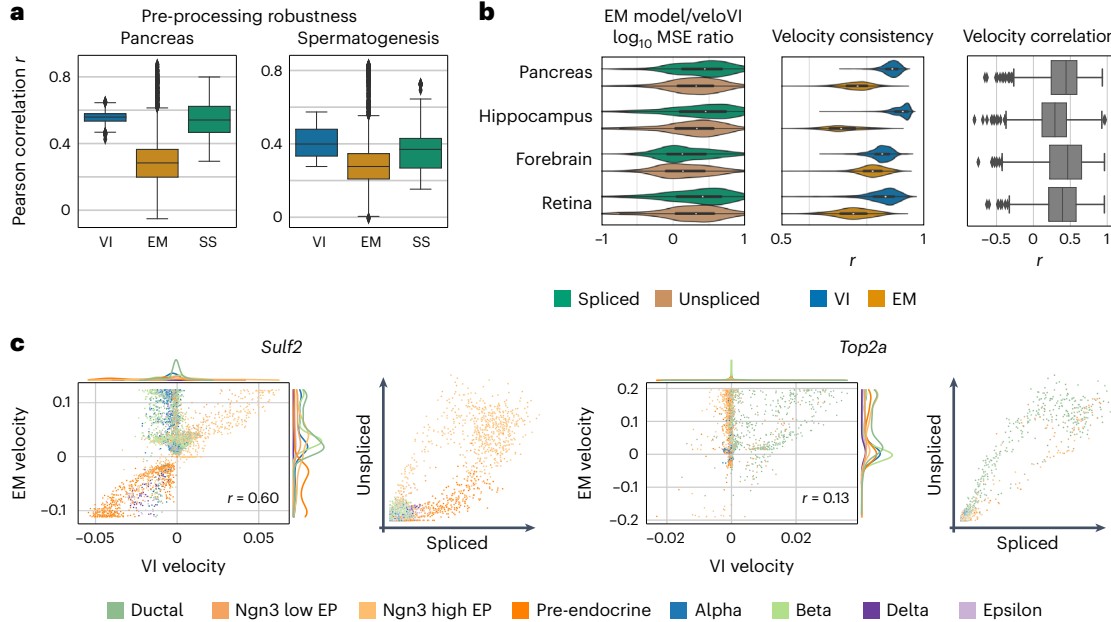

**Fig. 3 | Velocity model comparison in complex biological systems.**
**a**, Comparison of velocity estimation when using different algorithms to quantify unspliced and spliced counts. Correlation of velocities derived from pairs of quantification algorithms and from velocities estimating using one of veloVI (VI), the EM and steady-state (SS) model are compared on the pancreas[35] and spermatogenesis[36] data. Box plots indicate the median (center line), interquartile range (hinges) and whiskers at 1.5× interquartile range ($n = 78$ pairs of quantification methods each). **b**, Comparison of veloVI and the EM model based on gene-wise MSE (left), cell-wise velocity consistency (middle) and gene-wise velocity Pearson correlation (right) on datasets of pancreas endocrinogenesis[35]

($n = 1{,}074$ genes for MSE and velocity correlation; $n = 3{,}696$ cells for velocity consistency), hippocampus[8] ($n = 1{,}292$ genes for MSE and velocity correlation; $n = 18{,}213$ for velocity consistency), forebrain[8] ($n = 822$ genes for MSE and velocity correlation; $n = 1{,}720$ for velocity consistency) and retina[43] ($n = 700$ genes for MSE and velocity correlation; $n = 2{,}726$ for velocity consistency). Box plots indicate the median (center line), interquartile range (hinges) and whiskers at 1.5× interquartile range. **c**, Velocity comparison on the level of individual genes in the pancreas dataset (*Sulf2* and *Top2a*). For each gene, the velocity of the EM model is plotted against veloVI (left) and the gene phase portrait is given (right). Each observation is colored by its cell type as defined in previous work[35].

phenotypic directionality suggested by the velocity vector in each cell (here, intrinsic uncertainty) and (2) quantify the variability of predicted future cell states under the velocity-induced cell–cell transition matrix (here, extrinsic uncertainty; Fig. 1b and Methods).

We applied these uncertainty metrics to the pancreas dataset (Fig. 4a). We observed that the intrinsic uncertainty was elevated in ductal and Ngn3-low endocrine progenitor populations, while the extrinsic uncertainty highlighted these same populations in addition to terminal alpha and beta cells. These results demonstrate that lower intrinsic uncertainty does not necessarily preclude higher extrinsic uncertainty. While the former relies on estimating the velocity vector (which is cell-intrinsic), most velocity pipelines also account for other cells in the dataset, which presumably represent the potential past and future states of the cell, to determine cell transitions (Fig. 1b). In the case of alpha and beta cells, these cells represent terminal populations in the pancreas dataset, which may explain the high extrinsic uncertainty as there are no observed successor states. Conversely, in the case of transient cell populations, such as Ngn3-high endocrine progenitors and pre-endocrine cells, both metrics assign a low uncertainty. We attribute the low intrinsic uncertainty of these cells to the fact that their dynamics agree well with the underlying model assumptions (Extended Data Fig. 3). The addition of low extrinsic uncertainty further suggests that these cell types have clear successor populations in this dataset (Fig. 4a).

To further understand what aspects of the data these uncertainty metrics capture, we (in silico) perturbed the pancreas dataset by either (1) downsampling the total counts of each cell to mimic changes in sequencing depth and capture efficiency; (2) subsampling unspliced counts for a subset of genes to mimic the biased capture of unspliced molecules; or (3) adding random multiplicative noise to each abundance value (Methods). We applied each perturbation at various strengths and found that for each perturbation source, the intrinsic

uncertainty increased with the perturbation strength. We found a similar response for the extrinsic uncertainty except in the case of total count downsampling, which required a high strength to shift the extrinsic uncertainty (Extended Data Fig. 4). These results suggest that the uncertainty metrics can capture random noise in the data, as well as bias in how the transcripts are measured.

Finally, we asked whether we could use veloVI's uncertainty to address the common behavior of unexpected 'backflow' in two-dimensional velocity visualizations; when projecting the average veloVI velocity onto a Uniform Manifold Approximation and Projection (UMAP)[40] plot (using procedures from elsewhere[11]), we observed an incorrect 'backflow' of directionality in alpha and beta cells, which showed transitions toward their known progenitors. While these terminal populations have high extrinsic uncertainty according to veloVI, it remains difficult to explain which genes cause the inconsistency. In the case of scVelo, it has been proposed to use the likelihood of a gene as a proxy; however, the likelihood has no direct connection to cell–cell transition-based analyses.

To this end, we sought to score genes in each cell according to how well their velocity agrees with the predicted future cell state that is derived via the velocity-induced transition matrix (incorporating velocity information from all genes as well as gene expression in neighboring cells; Methods). We reasoned that this score, which we call velocity coherence, could help gain insight into why a particular directionality might manifest. A positive score of a gene indicates the velocity value of that gene (the time derivative of its spliced mRNA) agrees with its expression in the inferred future cell state (same direction) and likewise, a negative score indicates disagreement (Fig. 4b and Extended Data Fig. 5a).

In the alpha cells, for example, there are both positively and negatively scoring genes. Genes with a negative score, such as *Gcg* and

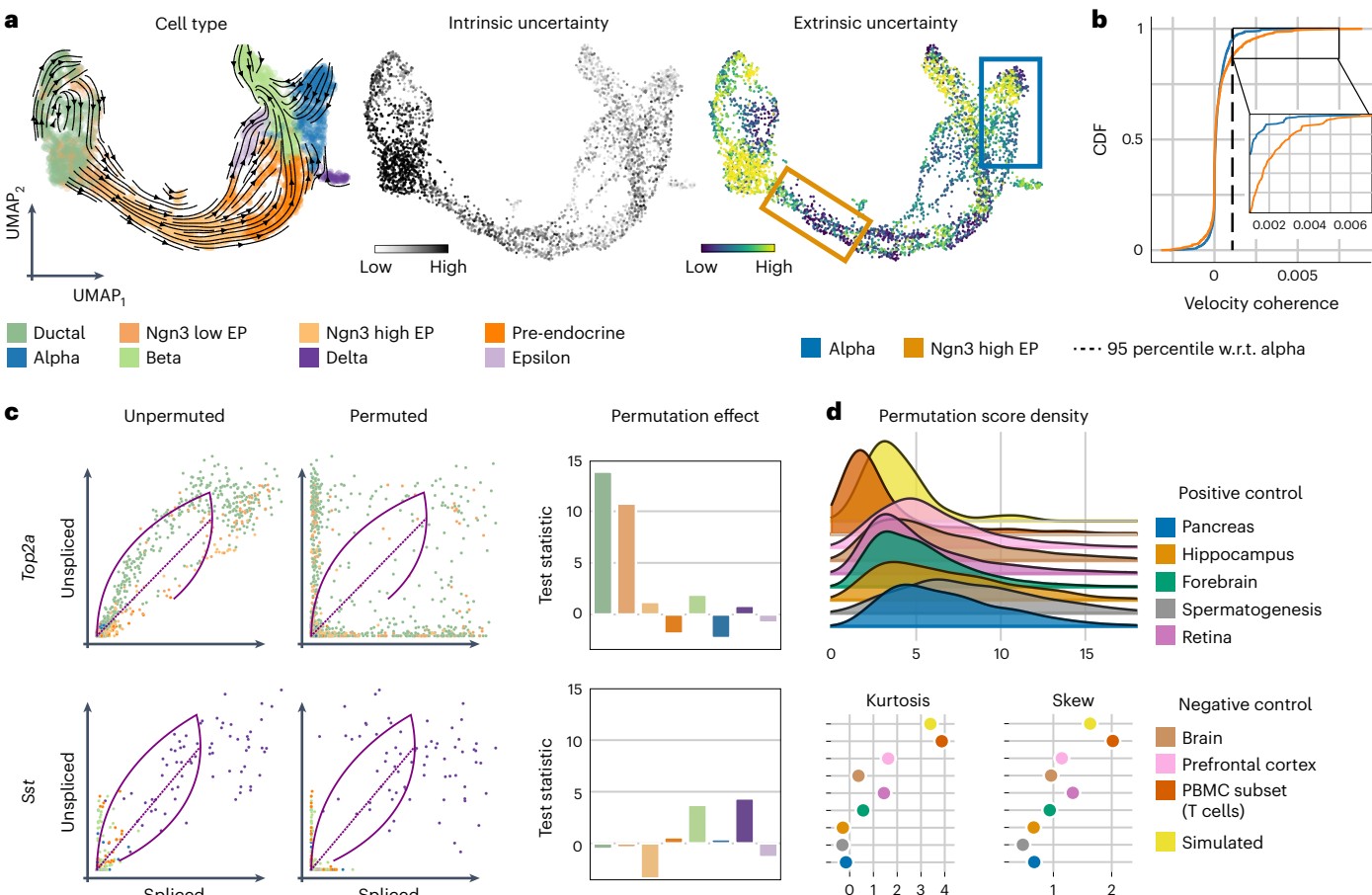

**Fig. 4 | Velocity uncertainty and permutation score analysis. a**, From left to right: velocity stream, intrinsic and extrinsic uncertainty of the pancreas dataset estimated by veloVI. The left UMAP embedding is colored by cell type according to original cluster annotations[11,35]. Uncertainty is defined as the variance of the cosine similarity between samples of the velocity vector (intrinsic) and future cell states (extrinsic) and their respective mean (Methods). **b**, The corresponding cumulative distribution functions (CDFs) of the gene velocity coherence score is shown for alpha and Ngn3-high EP cells. The velocity coherence, defined for one cell and gene as the product of the velocity and the expected displacement of that cell/gene, is averaged within cell types. w.r.t., with regard to. **c**, The effect on the error between inferred dynamics and data when permuting unspliced and spliced abundance in the case of *Top2a* (top) and *Sst* (bottom). Coloring of cell types is according to **a**. **d**, Permutation score densities of datasets of the pancreas[35], hippocampus[8], forebrain[8], spermatogenesis[36], retina[43], brain[39], prefrontal cortex[38], PBMCs and simulated data[15] (top). Kurtosis (left) and skew (right) for each dataset (bottom).

*Sphkap*, were fit correctly by veloVI (alpha cells after pre-endocrine cells in time along the inferred trajectory on the phase portrait), but disagree with the predicted future cell state, suggesting that other genes are outweighing these genes in the transition matrix computation (Extended Data Fig. 5b). Indeed, genes such as *Rnf130*, *Etv1* and *Grb10*, which had a positive score that agrees with the backflow, seemed to have been fit incorrectly (alpha cells precede pre-endocrine cells along the inferred trajectory on the phase portrait) (Extended Data Fig. 5c). The incorrect fits can putatively be explained by violated model assumptions such as a transcriptional burst in alpha cells (*Rnf130*), ambiguous phase portraits (*Etv1*) and multi-kinetics (*Grb10*).

Conversely, the dynamics in Ngn3-high cells are correctly visualized in the UMAP representation (Fig. 4a). We attribute this result to the presence of many genes agreeing with both the model assumptions and the predicted future state of a cell (Extended Data Fig. 5d). Compared to the 95% percentile of the coherence score in alpha cells, more than twice as many genes ranked above this threshold in the Ngn3-high cluster (135 versus 54); however, even in this case, we found that many genes were fit with incorrect dynamics for this cell type (Extended Data Fig. 5e).

Taken together, these results suggest that the visualization of dynamics on a two-dimensional embedding with previously described procedures is explained by small subsets of genes. Thus, caution is warranted when analyzing projections of velocity estimates onto a two-dimensional embedding of the data. We urge users to investigate the dynamics at the level of individual genes to identify which genes meet the model assumptions. Putative candidates are given by our proposed velocity coherence score. Additionally, to identify genes viable for RNA velocity analysis due to the presence of transient cell populations, we propose a score outlined next.

### veloVI identifies insufficiently observed or steady-state dynamics

In datasets with non-differentiating, hierarchically-related cell types, spurious cell state transitions may manifest when applying RNA velocity[14,15]. Indeed, the underlying transcriptional likelihood model cannot readily distinguish between the case of a transient population and that of multiple steady-state populations. Therefore, we devised a procedure to use a trained veloVI model to identify genes with phase portraits that are consistent with a developmental process versus ones that are consistent with steady-state dynamics or are confounded by noise.

We reasoned that the model fit of genes showing only steady-state dynamics would be robust to a permutation of the data while the model fit of genes with transient populations would worsen. Specifically for every gene, cell type and species (spliced/unspliced) independently,

we permuted the abundances of cells in a manner equivalent to shuffling cell barcodes. Subsequently, we passed this perturbed dataset through the veloVI model's trained encoder and decoder and recorded the absolute error of the fit grouped by genes and cell types. We then used the $t$-test statistic to compare the mean absolute error in each cell-type-gene group between the perturbed and original dataset (Extended Data Fig. 6 and Methods). We hypothesized that the $t$-test statistic, capturing the effect of the permutation, would be elevated in transient populations with strong time dependence and, conversely, near-zero in steady-state populations.

In the pancreas dataset, the permutation strongly affects the ductal and Ngn3 low EP cells for the cell-cycle gene *Top2a*. Indeed, these cell types trace fully-observed induction and repression states for *Top2a*. In the case of the delta-specific gene *Sst*, where no such transient connection is observed, for example from ductal to pre-endocrine to delta cells, no single cell type is strongly affected when permuting (Fig. 4c). Consequently, even though *Sst* is essential for the identity of delta cells, the gene does not display continuous dynamics from ductal progenitor cells and, thus, does not include the necessary information to be analyzed with RNA velocity.

We then applied this procedure to a variety of datasets. In one set of tests, we used datasets describing cellular development. These datasets serve as partial positive controls as we expect directed dynamical processes, as modeled by RNA velocity, to take place in at least a subset of cells in the dataset. As negative controls, we used simulated data of bursty kinetics[15] with no overall differentiation of cell state and datasets containing multiple cell types that are in steady-state. To summarize the permutation for a gene, we used the maximum permutation effect $t$-test statistic across cell types (permutation score). Two clusters of datasets emerged when characterizing the per-gene permutation score distribution (Fig. 4d). One cluster, with a fatter right tail (quantified by skewness and kurtosis), contained positive control datasets such as the pancreas and spermatogenesis. Despite having relatively many genes sensitive to permutation, the datasets of this cluster also contained many genes that were not sensitive, suggesting that there are likely many non-dynamical genes used for downstream analysis with RNA velocity. The other cluster, with less density in the right tail, contained negative controls such as the peripheral blood mononuclear cells (PBMCs), null-data simulation and the prefrontal cortex.

Between these two clusters of datasets, we also found a few ambiguous datasets, such as the mouse retina (positive control) and brain (negative control), which suggests that there exist some cell subsets within these datasets that are affected by the permutation and hence, possibly reflect a directed dynamical process that is appropriate for modeling with RNA velocity; however, upon closer inspection of the brain dataset, we identified mature neurons as responsible for skewing the permutation score density (Extended Data Fig. 7a). The cluster of mature neurons was singled out as it attributes for about one-third of the highest permutation scores (Extended Data Fig. 7b). For the genes with the highest permutation score, these neuronal cells exhibit a bimodal distribution in which one mode has low unspliced and spliced abundance while the other has respectively higher abundances (Extended Data Fig. 7c). Thus, we attribute this skewing to coarse labeling of this population (Extended Data Fig. 7d). When excluding mature neurons from this analysis, the distribution shifted and its key characteristics moved toward the cluster formed by the negative control cases (Extended Data Fig. 7e).

In the accompanying code to this manuscript, we provide these permutation score densities as a resource for users of RNA velocity, which will enable the datasets we analyzed here to serve as references for the score distribution and thus as a systematic approach to measure the overall transient dynamics of a dataset. For example, datasets exhibiting similar permutation score distributions as the given negative control cases (for example, via kurtosis or skew) are not suitable for RNA velocity analysis with current models.

In Supplementary Notes 1 and 2, we provide case studies outlining how veloVI can be used in practice on PBMCs (negative control) and mouse developing dentate gyrus (partial positive control). These demonstrations synthesize veloVI's uncertainty quantification and permutation procedure along with the velocity coherence. When applying the permutation procedure, we were able to provide further evidence for the lack of transient populations in the case of PBMCs (Supplementary Note 1), as well as identify transient populations of neuroblasts and granule immature cells for many genes in dentate gyrus (Supplementary Note 2). Taken together, these results demonstrate that the permutation score is also useful for identifying cell populations that lack detectable transient dynamics.

#### veloVI is an extensible framework for dynamical modeling

The transcriptional model assumptions at the level of one gene (for example, constant rates that impose a specific structure of phase portraits) can be shown to be violated in many cases. For example, in the case of transcriptional bursts in which the transcription rate increases with time[18] or multiple kinetics within a single gene[14], the assumption of constant kinetic rates is violated. Thus, there remains a need for modeling frameworks that are extensible and support varied and more nuanced dynamical assumptions. While veloVI makes many of the same assumptions as in the EM model, it leverages black-box computational and statistical techniques that allow its generative model to be altered to include new assumptions without needing to extensively rewrite inference recipes or generally sacrifice scalability.

To explore veloVI as a general modeling framework, we adapted it to use gene-specific, time-dependent transcription rates. Under this extension, transcription rates are free to monotonically increase or decrease with respect to time[14], thus allowing for modeling the acceleration of RNA abundance, which can impact the curvature of the model fit (Methods and Fig. 5a). To infer these additional parameters, only the likelihood function of veloVI needed to be adapted. Applying this modified version of veloVI to the pancreas, dentate gyrus and forebrain datasets, we observed improved fit for the majority of genes (Fig. 5b). In the case of the pancreas dataset, the added flexibility allowed veloVI to better fit genes that seem more linear in their phase portraits, for example, as it can reduce the curvature of the fitted dynamics (Fig. 5c).

In the case of *Smarca1*, the model using a constant transcription rate inferred a downregulation (repression) of alpha cells differentiating into their progenitor populations of pre-endocrine cell and ductal cells (Fig. 5c). Contrastingly, using a time-dependent transcription rate, the upregulation of ductal to pre-endocrine to alpha cells is inferred by the generalized model. Similar observations apply to *Atad2* and *Cdkn1a*. While the constant transcription rate model inferred the correct regulation type for *Ppp1r1a*, its generalized counterpart captures the underlying dynamics more accurately (Fig. 5c). Overall, for most genes, we observed a decreasing transcription rate over time (Supplementary Fig. 8).

Altogether, this exemplary model extension demonstrates the flexibility of veloVI's modeling approach. The flexibility allows us to quickly prototype extensions and infer additional parameters within a single, consistent framework. We, thus, expect future models to benefit from such flexibility.

### Discussion

Here, we reformulated the estimation of RNA velocity in a variational inference framework with veloVI. Our method compares favorably to previously proposed methods[8,11] and adds actionable metrics into downstream data analyses at the cell level via uncertainty quantification and at the level of a gene and dataset with the permutation score. We believe that veloVI will facilitate more systematic analyses with RNA velocity and help reduce the strong reliance on prior knowledge to guide whether results are sensible. As an example, our permutation score could be used to filter genes that are considered for further

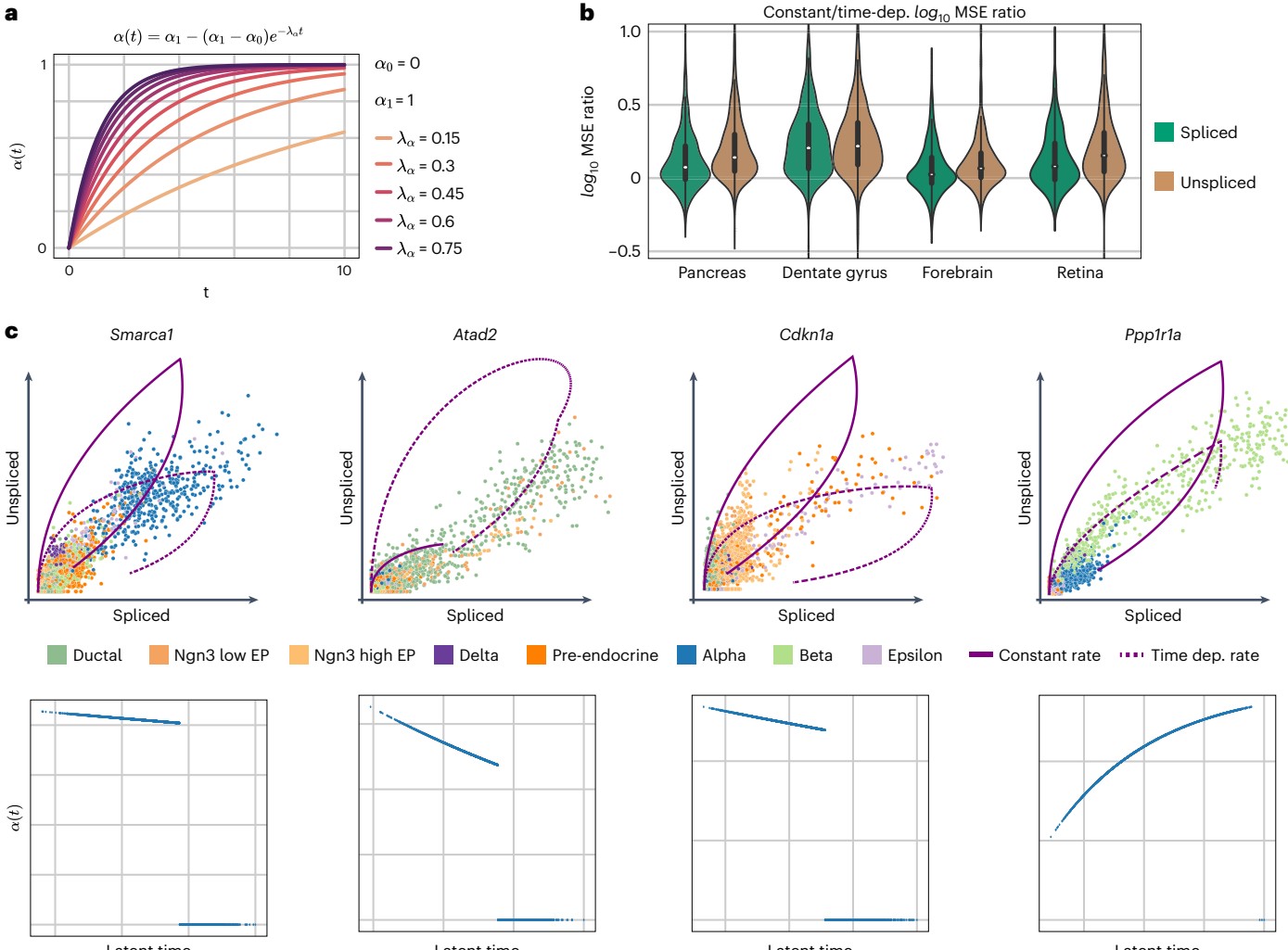

**Fig. 5 | Extension to modeling time-dependent transcription rates.**
**a**, Time-dependent transcription rate for different sets of parameter values.
**b**, Log$_{10}$ MSE ratio of models with constant and time-dependent transcription rate in the case of the pancreas ($n = 1,074$ genes), dentate gyrus ($n = 1,292$ genes), forebrain ($n = 822$ genes) and retina ($n = 700$ genes). Box plots indicate the median (center line) and interquartile range (hinges). **c**, Gene phase portrait with inferred dynamics with a constant (solid line) or time-dependent transcription rate (dashed line) (top). Corresponding time-dependent transcription rates are shown in terms of the inferred latent time (bottom).

analysis. We also note that related work has very recently incorporated deep learning with RNA velocity and we review these methods and compare them to veloVI in Supplementary Note 3.

We view this formulation of modeling transcriptional dynamics with probabilistic models and deep learning as a step toward a more rigorous pipeline that faithfully captures the biophysical phenomenon of RNA metabolism. In this work, we relied on previously described data processing approaches that smooth unspliced/spliced abundances across nearest neighbors before velocity estimation. We also borrowed many assumptions from the EM model, including, for example, the lack of explicit support for multiple diverging lineages that would result in genes reflecting a superposition of dynamical signals.

In contrast to previous models, veloVI is built in an extensible way using the scvi-tools framework[21]. As a proof of concept, we demonstrated that veloVI could be easily extended to use time-dependent transcription rates, which improved model fit for many genes. We anticipate that the veloVI framework will be further adapted to overcome other computational challenges including estimating velocity while accounting for batch effects, using multimodal technologies with measurements that span biology's central dogma[41,42] and directly modeling the unspliced and spliced RNA counts with count-based

likelihoods. Furthermore, while veloVI's estimated velocities are relative to a given maximum time of the process (similar as for the EM model), they are no longer relative with respect to the splicing rate as in the steady-state model. In future iterations, we anticipate including prior information from metabolic labeling data to estimate absolute velocities. We discuss these challenges, other considerations and future opportunities in Supplementary Note 4.

A philosophical challenge with RNA velocity relates to the notion that models should use bottom-up mechanistic approaches while also being general enough to be applied across a variety of biological systems, each with their own caveats and unique dynamics. In this work, we use a low-dimensional representation of a cell's phenotypic state to capture multiple biological processes (for example, differentiation and cell cycle). More complex models likely need prior information, such as known experimental time points or cell type lineages to solve issues of statistical identifiability that arise in these more general modeling scenarios; however, incorporating such priors can contradict the usage of RNA velocity as a de novo discovery tool for the trajectory inference task. Despite all these outlined challenges, we envision that veloVI will facilitate applications of RNA velocity via uncertainty-aware analysis as well as easier model prototyping, benefiting both users and method developers.

## Online content

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

## Methods

### veloVI model specification

We begin with the formulation of the 'dynamical' model of RNA velocity as presented by ref. [11]. We posit transcriptional states $k \in \{1, 2, 3, 4\}$, where $k = 1$ indicates induction, $k = 2$ indicates the induction steady state, $k = 3$ indicates repression and $k = 4$ indicates the repression steady state.

Let $\alpha_{gk}$ be the gene-state-specific reaction rate of transcription. Let $\beta_g$ be the gene-specific splicing rate constant and let $\gamma_g$ be the gene-specific degradation rate constant. Each gene has a switching time $t_g^s$ when the system switches from induction phase to repression phase.

Given the solution to the ordinary differential equations [11], the unspliced transcript abundance at time $t_{ng}$ for cell $n$ and gene $g$ is defined as

$$\bar{u}^{(g)}(t_{ng}, k) := u_{gk}^0 e^{-\beta_g(t_{ng} - t_{gk}^0)} + \frac{\alpha_{gk}}{\beta_g}\left(1 - e^{-\beta_g(t_{ng} - t_{gk}^0)}\right), \tag{1}$$

where $t_{gk}^0$ is the initial time of the system in state $k$. The spliced transcript abundance is defined as

$$\begin{aligned}\bar{s}^{(g)}(t_{ng}, k) :=& s_{gk}^0 e^{-\gamma_g \tau} + \frac{\alpha_{gk}}{\gamma_g}\left(1 - e^{-\gamma_g(t_{ng} - t_{gk}^0)}\right) \\ &+ \frac{\alpha_{gk} - \beta_g u_{gk}^0}{\gamma_g - \beta_g}\left(e^{-\gamma_g(t_{ng} - t_{gk}^0)} - e^{-\beta_g(t_{ng} - t_{gk}^0)}\right).\end{aligned} \tag{2}$$

**Induction state.** For the induction state, $k = 1$, we have $u_{g1}^0 = 0$, $s_{g1}^0 = 0$, $\alpha_{g1} > 0$ and $t_{g1}^0 = 0$. Thus, the unspliced transcript abundance can then be expressed as

$$\bar{u}^{(g)}(t_{ng}, k = 1) := \frac{\alpha_{g1}}{\beta_g}\left(1 - e^{-\beta_g t_{ng}}\right). \tag{3}$$

Likewise, the spliced transcript abundance can be simplified to

$$\bar{s}^{(g)}(t_{ng}, k = 1) := \frac{\alpha_{g1}}{\gamma_g}\left(1 - e^{-\gamma_g t_{ng}}\right) + \frac{\alpha_{g1}}{\gamma_g - \beta_g}\left(e^{-\gamma_g t_{ng}} - e^{-\beta_g t_{ng}}\right). \tag{4}$$

**Induction steady state.** For the induction steady state, $k = 2$, the unspliced and spliced transcript abundances are defined as limits of the system:

$$\bar{u}^{(g)}(t_{ng}, k = 1) := \lim_{t_{ng} \to \infty} \bar{u}^{(g)}(t_{ng}, k = 1) = \frac{\alpha_{g1}}{\beta_g} \tag{5}$$

$$\bar{s}^{(g)}(t_{ng}, k = 2) := \lim_{t_{ng} \to \infty} \bar{s}^{(g)}(t_{sg}, k = 1) = \frac{\alpha_{g1}}{\gamma_g}. \tag{6}$$

**Repression state.** For the repression state, $k = 3$, we have $\alpha_{g3} = 0$ and $t_{g3}^0 = t_g^s$. Thus, the number of unspliced transcripts can then be expressed as

$$\bar{u}^{(g)}(t_{ng}, k = 3) := u_{g3}^0 e^{-\beta_g(t_{ng} - t_{g3}^0)}. \tag{7}$$

Likewise, the number of spliced transcripts can be simplified to

$$\bar{s}^{(g)}(t_{ng}, k = 3) := s_{g3}^0 e^{-\gamma_g(t_{ng} - t_{g3}^0)} - \frac{\beta_g u_{g3}^0}{\gamma_g - \beta_g}\left(e^{-\gamma_g \tau} - e^{-\beta_g(t_{ng} - t_{g3}^0)}\right). \tag{8}$$

The initial conditions, $u_{g3}^0$ and $s_{g3}^0$ are defined by the induction model at the switching time $t_g^s$, such that

$$u_{g3}^0 = \bar{u}^{(g)}(t_{sg}, k = 2) \tag{9}$$

$$s_{g3}^0 = \bar{s}^{(g)}(t_{sg}, k = 2). \tag{10}$$

**Repression steady state.** For the repression steady state, the limit upon which $t_{ng} \to \infty$, there is no expression, so we have

$$\bar{u}^{(g)}(t_{ng}, k = 4) := 0 \tag{11}$$

$$\bar{s}^{(g)}(t_{ng}, k = 4) := 0. \tag{12}$$

**Model assumptions.** As in ref. [11], this model assumes that for one gene, at the initial time of the system, cells are first in induction phase in which both spliced and unspliced expression increases. Then cells potentially reach a steady state of this induction state. Next at some future time $t_g^s$ the system switches to repression state. Finally, the repression reaches a steady state in which there is no expression. Further assumptions are necessary to identify the dynamical model parameters [44]; thus, we assume that each gene is on the same time scale (precisely each gene has a maximum time of $t = 20$ as shown previously [11]).

### veloVI generative process

We posit a generative process that takes into account the underlying dynamics of the system. Compared to Bergen et al. [11], the model here does not treat each gene independently; instead, the latent time and states for each (cell and gene) pair are tied together via a local low-dimensional latent variable.

For each cell we draw a low-dimensional ($d = 10$ dimensions throughout this manuscript) latent variable

$$z_n \sim \text{Normal}(0, I_d) \tag{13}$$

that summarizes the latent state of each cell. Next, for each gene $g$ in cell $n$ we draw the distribution over the state assignments as well as the state assignment itself

$$\pi_{ng} \sim \text{Dirichlet}(0.25, 0.25, 0.25, 0.25) \tag{14}$$

$$k_{ng} \sim \text{Categorical}(\pi_{ng}) \tag{15}$$

Here $\pi_{ng}$ is sampled from a Dirichlet distribution, which has the support of the probability simplex. In other words, the Dirichlet provides a distribution over discrete probability distributions. If $k_{ng} = 1$ (induction), then the time is a function of $z_n$,

$$\rho_{ng}^{(1)} = [h_{\text{ind}}(z_n)]_g \tag{16}$$

$$t_{ng}^{(1)} = \rho_{ng}^{(1)} t_g^s \tag{17}$$

where $h_{\text{ind}} : \mathbb{R}^d \to (0, 1)^G$ is parameterized as a fully connected neural network. Notably, this parameterization results in an induction-specific time that is constrained to be less than the switching time.

Else, if $k_{ng} = 3$ (repression),

$$\rho_{ng}^{(3)} = [h_{\text{rep}}(z_n)]_g \tag{18}$$

$$t_{ng}^{(3)} = \left(t_{\max} - t_g^s\right)\rho_{ng}^{(3)} + t_g^s \tag{19}$$

where $t_{\max} := 20$ is used to fix the time scale across genes and identify the rate parameters of the model. Similarly to the previously defined function, $h_{\text{rep}} : \mathbb{R}^d \to (0, 1)^G$ and is also a neural network.

We also consider two potential steady states. If $k_{ng} = 2$ (induction steady state) or if $k_{ng} = 4$ (repression steady state), we consider the limit as time approaches $\infty$, which is described in the previous section.

Finally, the observed data are sampled from normal distributions as

$$u_{ng} \sim \text{Normal}\left(\bar{u}^{(g)}(t_{ng}^{(k_{ng})}, k_{ng}), (c_k \sigma_g^u)^2\right) \quad (20)$$

$$s_{ng} \sim \text{Normal}\left(\bar{s}^{(g)}(t_{ng}^{(k_{ng})}, k_{ng}), (c_k \sigma_g^s)^2\right) \quad (21)$$

For veloVI, we consider the observed data $\{(s_n, u_n)\}_{n=1}^N$ to be the nearest-neighbor smoothed expression data that is also used as input to scVelo as well as velocyto. In addition, we assume the data have been preprocessed such that for each gene, the smoothed spliced and unspliced abundances are independently min-max scaled into [0, 1]. By using the normal distribution, we assume that the smoothed expression (which represents an average of random variables) has a sampling distribution centered on some mean value and that this sampling distribution is approximately normal; however, the flexibility of this modeling framework will enable extensions that consider the discrete nature of unique molecular identifiers used in standard scRNA-seq assays.

We include a state-dependent scaling factor on the variance. For all experiments in this manuscript, we used $c_k = 1$ except for the repression steady state in which $c_4 = 0.1$. This hyperparameter choice forces the variance of abundance in the repression steady state to be less than that of other transcriptional states, which reflects the notion that the repression steady state corresponds to zero transcriptional activity. Despite the assumption of zero transcriptional activity, the normal distribution here captures noise that arises during the experimental process (ambient transcripts) as well as during preprocessing (for example, KNN smoothing). Finally, in the following, let $\theta$ be the set of parameters of the generative process ($\alpha$, $\beta$, $\gamma$, $t^s$ and neural network parameters).

## veloVI inference procedure

We seek the following: (1) point estimates of the transcription rate, degradation and splicing rate constants and the switching time point; (2) point estimates of the parameters of the neural networks; and (3), a posterior distribution over the latent variables, which in this case includes $z$ and $\pi$. Noting that the model evidence $p_\theta(u, s)$ cannot be computed in closed form, we use variational inference[45] to approximate the posterior distribution as well as accomplish the other tasks. Following inference, velocity can be calculated as a functional of the variational posterior distribution.

**Variational posterior.** We posit the following factorization on the approximate posterior distribution

$$q_\phi(z, \pi \mid u, s) := \prod_n^N q_\phi(z_n \mid u_n, s_n) \prod_g^G q_\phi(\pi_{ng} \mid z_n), \quad (22)$$

in which dependencies are specified using neural networks with parameter set $\phi$. Here $z$ factorizes over all $n$ cells and $\pi_{ng}$ over all $n$ cells and $g$ genes.

For the likelihoods, we integrate over the choice of transcriptional state $k_{ng}$, such that the likelihoods for unspliced and spliced transcript abundances,

$$p_\theta(u_{ng} \mid z_n, \pi_n) = \sum_{k_{ng} \in \{1,2,3,4\}} \pi_{ngk_{ng}} \text{Normal}\left(\bar{u}^{(g)}(t_{ng}^{(k_{ng})}, k_{ng}), (c_k \sigma_g^u)^2\right) \quad (23)$$

$$p_\theta(s_{ng} \mid z_n, \pi_n) = \sum_{k_{ng} \in \{1,2,3,4\}} \pi_{ngk_{ng}} \text{Normal}\left(\bar{s}^{(g)}(t_{ng}^{(k_{ng})}, k_{ng}), (c_k \sigma_g^s)^2\right) \quad (24)$$

are mixtures of normal distributions.

**Objective.** The objective that is minimized during inference is composed of two terms

$$\mathcal{L}_{\text{velo}}(\theta, \phi; u, s) = \mathcal{L}_{\text{elbo}}(\theta, \phi; u, s) + \lambda \mathcal{L}_{\text{switch}}(\theta; u, s), \quad (25)$$

where $\mathcal{L}_{\text{elbo}}$ is the negative evidence lower bound[45] of $\log p_\theta(u, s)$ and $\mathcal{L}_{\text{switch}}$ is an additional penalty that regularizes the location of the transcriptional switch in the phase portrait. In more detail,

$$\mathcal{L}_{\text{elbo}}(\theta, \phi; u, s) = \sum_n -\mathbb{E}_{q_\phi(z_n, \pi_n \mid u_n, s_n)} \left[ \log p_\theta(u_n, s_n \mid z_n, \pi_n) \right]$$
$$+ \text{KL}\left( q_\phi(z_n \mid u_n, s_n) \parallel p(z) \right)$$
$$+ \mathbb{E}_{q_\phi(z_n \mid u_n, s_n)} \left[ \sum_g \text{KL}\left( q_\phi(\pi_{ng} \mid z_n) \parallel p(\pi_{ng}) \right) \right], \quad (26)$$

which can be estimated using minibatches of data. In particular, we use randomly sampled minibatches of 256 cells for inference. For the penalty term $\mathcal{L}_{\text{switch}}$, we start by only considering cells that are above the 99th percentile of unspliced abundance for each gene. Using these cells we compute the median unspliced and spliced abundance for each gene separately. Let $u^*$ and $s^*$ be the outcome of this procedure, then

$$\mathcal{L}_{\text{switch}}(\theta; u, s) = \sum_g \left( u_{g3}^0 - u_g^* \right)^2 + \left( s_{g3}^0 - s_g^* \right)^2, \quad (27)$$

where $u_{g3}^0$ and $s_{g3}^0$ were defined as the initial conditions of the repression phase at the switch time $t_g^s$.

**Initialization.** We initialize $\alpha_{g1}$ to be equal to the median unspliced abundance for the cells above the 99th percentile for each gene. The other global parameters, including the splicing, degradation and switch time are initialized to a constant value shared by all genes. All neural network initialization uses the default implementation in PyTorch.

**Optimization.** To optimize $\mathcal{L}_{\text{velo}}$ we use stochastic gradients[26] along with the Adam optimizer with weight decay[46] as implemented in PyTorch[47]. For all experiments we use $\lambda = 0.2$ for scaling the regularization term in the loss. As a result of minibatching, veloVI's memory usage is constant throughout training. Unless otherwise specified, all neural networks are fully connected feedforward networks that use standard activation functions such as ReLU for hidden layers and softplus or exponential for parameterizing non-negative distributional parameters.

**Architecture.** An overview of the veloVI architecture is shown in Supplementary Fig. 9.

## Downstream tasks

**Fitted abundance values.** The fitted values (used, for example, in MSE benchmarks) for unspliced and spliced abundance are the posterior predictive mean:

$$\mathbb{E}_{p(u_n^* \mid u_n, s_n)} \left[ u_n^* \right], \, \mathbb{E}_{p(s_n^* \mid u_n, s_n)} \left[ s_n^* \right],$$

where $u_n^*$ and $s_n^*$ are unobserved random variables representing posterior predictive values of unspliced and spliced abundances for cell $n$. The posterior predictive in the case of unspliced abundance is defined as

$$p(u_n^* \mid u_n, s_n) = \int p_\theta(u_n^* \mid z_n, \pi_n) q_\phi(z_n, \pi_n \mid u_n, s_n) d\pi_n dz_n, \quad (28)$$

which uses the variational posterior distribution as a plug-in estimator for the true (unknown) posterior distribution.

We compare these fitted abundance values from veloVI to the analog of the EM model, which itself can be interpreted as a posterior predictive mean. Considering just the unspliced values, for example,

the EM model posits a normal likelihood $p(u_{ng}|t_{ng}, k_{ng})$ similar to veloVI but without the latent cell state $z_n$ and learns posterior distributions $q(t_{ng}|u_{ng}, s_{ng})$ and $q(k_{ng}|u_{ng}, s_{ng})$. Under the EM model, the posterior distributions are Dirac delta distributions and the corresponding posterior predictive is expressed as

$$p(u_{ng}^* | u_{ng}, s_{ng}) = \int p_\theta(u_{ng}^* | t_{ng}, k_{ng}) q_\phi(t_{ng}, k_{ng} | u_{ng}, s_{ng}) dk_{ng} dt_{ng}. \quad (29)$$

**State assignment.** The state assignment for each gene and cell is the approximate posterior mean

$$\mathbb{E}_{q_\phi(z_n|u_n, s_n)}\left[\mathbb{E}_{q_\phi(\pi_{ng}|z_n)}[\pi_{ng}]\right].$$

**Gene-wise latent time.** The latent time is computed for each gene and cell as

$$\mathbb{E}_{q_\phi(z_n|u_n, s_n)}\left[\mathbb{E}_{q_\phi(\pi_{ng}|z_n)}\left[t_{ng}^{(k_{ng})}\right]\right],$$

where the outer expectation with respect to $q_\phi(z_n|u_n, s_n)$ is estimated with Monte Carlo samples, while the inner expectation is computed analytically over the transcriptional states $k_{ng}$.

**RNA velocity.** The velocity of a particular gene in a particular cell is similarly a function of the variational posterior. Recall that the velocity is computed as

$$v^{(g)}(t^{(k)}, k) := \left.\frac{d\bar{s}^{(g)}(t, k)}{dt}\right|_{t^{(k)}} = \beta_g \bar{u}^{(g)}(t^{(k)}, k) - \gamma_g \bar{s}^{(g)}(t^{(k)}, k).$$

Thus, we can compute samples of a posterior predictive velocity distribution via the following process

1. Sample $z_n$ from $q_\phi(z_n|u_n, s_n)$.
2. Compute $\mathbb{E}_{q_\phi(\pi_{ng}|z_n)}\left[v^{(g)}\left(t_{ng}^{(k_{ng})}, k_{ng}\right)\right]$ for each gene.

   This provides samples from a distribution over the velocity for every gene–cell pair, which we then use in downstream tasks.

**Intrinsic uncertainty.** Let $\bar{v}_n$ be the posterior predictive velocity mean from the procedure above. The intrinsic uncertainty is then computed as $\mathbb{V}\text{ar}_{q_\phi(v_n|u_n, s_n)}[c(v_n, \bar{v}_n)]$ where $c$ denotes the cosine similarity. In effect, denote by $\{v_n^{(l)}\}_{l=1}^L$ the set of $L$ velocity vector samples of cell $n$ from the variational posterior. Then we have:

$$\hat{\sigma}_n^2 = \frac{1}{L-1} \sum_{l=1}^L \left(\frac{v_n^{(l)} \cdot \bar{v}_n}{\| v_n^{(l)} \| \| \bar{v}_n \|} - \frac{1}{L} \sum_{j=1}^L \frac{v_n^{(j)} \cdot \bar{v}_n}{\| v_n^{(j)} \| \| \bar{v}_n \|}\right)^2. \quad (30)$$

In this manuscript, we use $L = 100$ samples.

**Extrinsic uncertainty.** Let $T(v_{1:N}, s_{1:N})$ be a function that maps the velocity vectors and spliced abundances of the entire dataset (with $n$ cells) to a cell–cell transition matrix computed as described previously[11]. Namely, this function compares the similarity of the displacement $\delta_{ij}$ of nearest neighbors $s_i$ and $s_j$ (defined using $s_{1:N}$) to the velocity of cell $i$, $v_i$, via the cosine similarity

$$\cos(\delta_{ij}, v_i) = \frac{\delta_{ij}^T v_i}{\| \delta_{ij} \| \| v_i \|} \quad (31)$$

as the basis for computing transition probabilities between pairs of cells.

Following the construction of $T(v_{1:N}, s_{1:N})$ for one sample of velocity, the predicted future cell state is computed by the matrix multiplication $T(v_{1:N}, s_{1:N})S$, where $S$ is the cells by genes matrix of spliced RNA

abundances. These predicted future cell state vectors (over samples of velocity) then undergo the same variance computation procedure as described for the intrinsic uncertainty (namely, variance of the cosine similarity).

**Time-dependent transcription rate**
To highlight veloVI's extensibility with respect to model choice, we consider the time-dependent transcription rate

$$\alpha^{(k)}(t) = \begin{cases} \alpha_1 - (\alpha_1 - \alpha_0)e^{-\lambda_\alpha t}, & k \in \{1, 2\}, \\ 0, & k \in \{3, 4\}, \end{cases} \quad (32)$$

with parameters $\alpha_0, \alpha_1, \lambda_\alpha \in \mathbb{R}^+$ and $k$ indicating the transcriptional state. The system of differential equations describing the process of splicing stays otherwise unchanged and is, thus, given by

$$\begin{aligned} \dot{u} &= \alpha^{(k)}(t) - \beta u \\ \dot{s} &= \beta u - \gamma s. \end{aligned} \quad (33)$$

Consequently, it is of the general form

$$\dot{x} = Ax + g(t), \quad (34)$$

with dependent variable $x$, system matrix $A$, inhomogeneity $g(t)$ and solution

$$x(t) = x_0 e^{A(t-t_0)} + e^{At} \int_{t_0}^t e^{-As} g(s) ds. \quad (35)$$

As the abundance of unspliced mRNA is modeled independently of its spliced counterpart, its solution of equation (33) can be found directly. Comparing equation (33) with equations (34) and (35), we find that $x = u, A = -\beta, g(t) = \alpha^{(k)}(t)$. Consequently, the abundance of unspliced mRNA at time $t$ is given by

$$\begin{aligned} u(t) &= u_0^{(k)} e^{-\beta\tau^{(k)}} + \alpha_1^{(k)} e^{-\beta t} \int_{t_0^{(k)}}^t e^{\beta s} ds - \left(\alpha_1^{(k)} - \alpha_0^{(k)}\right) e^{-\beta t} \int_{t_0^{(k)}}^t e^{\beta s} e^{-\lambda_\alpha^{(k)} s} ds \\ &= u_0^{(k)} e^{-\beta\tau^{(k)}} + \frac{\alpha_1^{(k)}}{\beta}\left(1 - e^{-\beta\tau^{(k)}}\right) \\ &\quad - \frac{\alpha_1^{(k)} - \alpha_0^{(k)}}{\beta - \lambda_\alpha^{(k)}} e^{-\lambda_\alpha^{(k)} t_0^{(k)}} \left(e^{-\lambda_\alpha^{(k)}\tau^{(k)}} - e^{-\beta\tau^{(k)}}\right), \end{aligned} \quad (36)$$

with state-dependent initial time $t_0^{(k)}, \tau^{(k)} = t - t_0^{(k)}$ and $u_0^{(k)} = u(t_0^{(k)})$.

Similarly, this allows solving for $s(t)$, with $x = s, A = -\gamma, g(t) = \beta u(t)$. Applying solution formula (35), the abundance of spliced mRNA at time $t$ is given by

$$\begin{aligned} s(t) &= s_0^{(k)} e^{-\gamma\tau^{(k)}} + e^{-\gamma t} \int_{t_0^{(k)}}^t e^{\gamma t'} \beta u(t') dt' \\ &= s_0^{(k)} e^{-\gamma\tau^{(k)}} + \frac{\alpha_1^{(k)}}{\gamma}\left(1 - e^{-\gamma\tau^{(k)}}\right) + \frac{\alpha_1^{(k)} - \beta u_0^{(k)}}{\gamma - \beta}\left(e^{-\gamma\tau^{(k)}} - e^{-\beta\tau^{(k)}}\right) \\ &\quad - \frac{\beta\left(\alpha_1^{(k)} - \alpha_0^{(k)}\right)}{\left(\beta - \lambda_\alpha^{(k)}\right)\left(\gamma - \lambda_\alpha^{(k)}\right)} e^{-\lambda_\alpha^{(k)} t_0^{(k)}} \left(e^{-\lambda_\alpha^{(k)}\tau^{(k)}} - e^{-\gamma\tau^{(k)}}\right) \\ &\quad + \frac{\beta\left(\alpha_1^{(k)} - \alpha_0^{(k)}\right)}{\left(\beta - \lambda_\alpha^{(k)}\right)(\gamma - \beta)} e^{-\lambda_\alpha^{(k)} t_0^{(k)}} \left(e^{-\beta\tau^{(k)}} - e^{-\gamma\tau^{(k)}}\right), \end{aligned} \quad (37)$$

These new functions can be used as the mean in the veloVI likelihood, thus allowing optimization in a similar manner as described previously, with the addition of the new parameters $\alpha_0, \alpha_1, \lambda_\alpha \in \mathbb{R}^+$.

**Data preprocessing**
All datasets were pre-processed following the same steps. Genes with fewer than 20 unspliced or spliced counts were removed.

Transcriptomic counts of each cell were normalized by their median, pre-filtered library size and the 2,000 most highly variable genes selected based on dispersion. The aforementioned steps are performed using scVelo's[11] `filter_and_normalize` function.

Following gene filtering and count normalization, the first 30 principal components were calculated and a nearest neighbor graph with $k = 30$ neighbors was constructed. In a final step, counts were smoothed by the mean expression across their neighbors to compute final RNA abundances. These steps were performed by scVelo's `moments` function.

To estimate RNA velocity, the preprocessed unspliced and spliced abundances were (gene-wise) min−max scaled to the unit interval. Following, the steady-state model was applied to the entire dataset. Genes for which the estimated steady-state ratio and $R^2$ statistic are positive were considered for further analysis. If not stated otherwise, this subset of genes was used for parameter inference of veloVI and the EM model.

All datasets used, with the exception of the PBMC dataset, were obtained with spliced and unspliced RNA quantification and details can be obtained from the original publication (Supplementary Table 1). In the case of the PBMC dataset, we quantified RNA abundances using the kallisto bustools RNA velocity workflow[28], using an index and defaults as described in the tutorial on the software's website and automatically annotated via totalVI[48] using the Seurat v.3 CITE-seq PBMC dataset[49,50] as a reference.

### Benchmarking against EM and steady-state models

VeloVI was benchmarked against the EM and steady-state model by first comparing the accuracy of inferred parameters on simulated data. For each number of observations (1,000, 2,000, 3,000, 4,000 and 5,000), we simulated ten datasets of unspliced and spliced counts with 1,000 kinetic parameter tuples (transcription rate $\alpha_g$, splicing rate $\beta_g$, degradation rate $\gamma_g$) following a multivariate log-normal distribution. Latent time is Poisson distributed with a maximum of 20 h with the switch from induction to transcription, $t_g^s$, taking place after 2–10 h. The simulations were performed using the `simulation` function as implemented in scVelo[11] with `noise_level=0.8`.

As an additional validation, we inferred kinetic rates for the pancreas data using both veloVI and the EM model. Following, we randomly sampled overall 2,000 estimated parameter tuples (transcription rate $\alpha_g$, splicing rate $\beta_g$, degradation rate $\gamma_g$, switch time $t_g^s$) from the union of the parameters estimated by either algorithm and simulated splicing kinetics with `noise_level=1`. As the data are simulated and rate parameters and time are known, the ground truth velocities are defined as well. For each model, the Spearman correlations between ground truth and inferred latent time were compared. We used the Spearman correlation as it is an order statistic. Contrastingly, in the case of velocity estimates, we relied on Pearson correlation.

To compare the runtimes of veloVI and EM model were run on random subsets a mouse retina dataset[12] containing 1,000, 3,000, 5,000, 7,500, 10,000, 15,000 and 20,000 cells. The EM model was run on an Intel(R) Core i9-10900K CPU @ 3.70 GHz CPU using eight cores. VeloVI was run on an Nvidia RTX3090 GPU.

In the case of real-world data, for each gene, we compared the MSE between the observed abundance and the model-predicted abundance. We did this for each of the veloVI and EM models and separately for spliced and unspliced abundances. The result is the MSE per gene, per method and per species. In the case of the EM model, the abundance prediction is directly a function of the rates, time and transcriptional state and in the case of veloVI, this is the posterior predictive mean. Additionally, for each gene, velocity estimates from the veloVI and EM models were compared through Pearson correlation.

In addition to the MSE, the model-specific velocity consistency[11] was also compared. The velocity consistency $c$ quantifies the mean Pearson correlation of the velocity $v(x_j)$ of a reference cell $x_j$ with the velocities of its neighbors $\mathcal{N}_k(x_j)$ in a KNN graph.

$$c = \frac{1}{k} \sum_{x \in \mathcal{N}_k(x_j)} \mathrm{corr}\,(v(x_j), v(x))$$

To calculate the consistency, we rely on scVelo's `velocity_confidence` function. This evaluation metric makes the assumption that better local consistency is inherently good, reflecting smooth changes in velocity over the phenotypic manifold. We note that this is a heuristic evaluation and the validity of this metric can be affected by, for example, low density of similar cell states, misspecification of the KNN graph due to only considering spliced RNA, etc.

If a 'ground truth' cellular ordering, for example, a cell-cycle score[13,29], is given, we can make use of this source of information to estimate 'ground truth' velocities $\hat{v}$ via finite differences. We estimated this heuristic by first taking the median per gene of the first-order moment smoothed, spliced RNA abundance of all cells at a given cell-cycle position $p_i$, which we denote by $\bar{s}^{(i)}$. Then, assuming the $p_i$ are ordered ($p_i < p_{i+1}$), $\hat{v}^{(i)}$ is defined as

$$\hat{v}^{(i)} \propto \bar{s}^{(i+1)} - \bar{s}^{(i)} \tag{38}$$

Finally, we compared the sign of all ground truth velocities with their inferred counterparts of veloVI and the EM model (which are aggregated per position in the same way) by computing the sign accuracy per gene. The sign accuracy, which is the fraction of times that the signs agree, accounts for positive velocity, negative velocity and zero velocity. As a baseline, we included a random predictor that chose positive, negative or zero velocity with equal probability. The scEU-seq cell-cycle data (RPE1-FUCCI cells)[29] included, on average, 9.63 (s.d. 7.01) observations per cell cycle position and the U2OS-FUCCI[13] dataset provided 1.15 (s.d. 0.36) observations per cell cycle position. In the case of the U2OS-FUCCI dataset, the ground truth ordering was derived by the original authors using a polar regression on the scatter-plot of the two FUCCI protein markers. In the case of the RPE1-FUCCI cells, the ground-truth ordering was derived by the original authors using a pseudotime method on the FUCCI protein marker values.

As an additional validation, for each gene, we fitted a GAM to the inferred velocities of the two models versus the cell-cycle score. Similarly to ref. 13, we transformed the cell-cycle score in each dataset to $I = [0, 2\pi]$. To take the periodic nature of the cell cycle into account, we fitted the GAM per gene using spliced RNA abundance $s_{ng}$ as the response and the score as the variable, where the cell-cycle score was transformed to the range $[I - 2\pi, I, 2\pi]$. For each gene, a GAM with a univariate spline term for the triple of (shifted) cell-cycle positions was fitted. For each feature, 20 splines of degree three were used. For each gene, we reported the $R^2$ score.

### Stability analysis across quantification algorithms

To assess the robustness of estimation using different means of quantifying unspliced and spliced reads, we relied on previously preprocessed and published data[34]. The collection contains outputs of variants of the alevin[32], kallisto/bustools[28] velocyto[8], dropEST[31] and starsolo[51] pipelines. For details of how the data were generated, we refer to the original work[34].

To compare estimation across quantification algorithms, we first defined a reference set of genes for which to calculate RNA velocity. The set of reference genes was defined as the set of genes kept by preprocessing the data of one quantification method. In the case of the dentate gyrus data, starsolo was chosen for the quantification method, for all others velocyto. Data were pre-processed according to our described pre-processing pipeline. Counts from all other quantification approaches the same pre-processing steps were followed, except for gene filtering. To prevent the reference genes from being filtered out, they are passed to the `filter_and_normalize` function via the argument `retain_genes`.

Velocities were estimated for the steady-state model, EM model and veloVI. The velocities of the first two models were quantified using the function `velocity` with `mode='deterministic'` and `mode='dynamics'`, respectively, implemented in scVelo[11]. For veloVI, model parameters were inferred using default parameters and mean velocities estimated from 25 samples drawn from the posterior.

To compare estimates across quantification algorithms, for each model, cell and pair of quantification algorithms, the Pearson correlation between the paired velocity estimates, was calculated. For each model, the correlation scores were aggregated by taking the mean over cells for one quantification algorithm pair to assess robustness. The distribution of this mean correlation over all quantification algorithm pairs is used for visualization in Fig. 4 and Extended Data Fig. 2.

### Analysis with uncertainty quantification and velocity coherence

We used extrapolated future states $Ts_n$ of a cell to evaluate if inferred velocities are coherent. The velocity $v_n$ of a given cell $n$ is coherent if it points in the same direction as the empirical displacement $\delta_n = Ts_n - s_n$. Directionality is compared by calculating the Hadamard product $\delta_n \circ v_n$. In case both vectors point in the same direction for a given cell, the resulting entry will be positive and negative otherwise. To aggregate the score we report its mean per gene and cell type.

To benchmark the uncertainty quantification, we started with the pancreas dataset and added one of three kinds of perturbations at various strengths. After applying each perturbation, we ran the standard veloVI pipeline and recorded the uncertainty metrics. The first perturbation consisted of downsampling the cells to X% of their original library size (thus removing $(1 − X)$% of their transcripts; and for unspliced and spliced separately). This was achieved with `scanpy.pp.downsample_counts`. The second perturbation consisted of binomial thinning of the unspliced counts with probability $P$ (`unspliced=np.random.binomial(unspliced, 1−P)`). The final perturbation was multiplicative random noise. To each spliced and unspliced abundance value (this time after library size normalization) we multiplied the value with lognormally distributed noise (`np.exp(np.random.normal(0, scale))`). Across all perturbations we used a common gene set that was derived from the standard veloVI pipeline; this ensures that the uncertainty values are comparable as they incorporate information across all genes.

### Permutation scoring

To quantify how robust the inferred dynamics are with respect to random permutations in the input data, we define a gene- and cell-type-specific permutation effect, which is then aggregated to a gene-specific permutation score (Extended Data Fig. 6). For this analysis, we considered all highly variable genes and did not filter our genes based on estimates of the steady-state model.

To calculate the score, the unspliced and spliced abundances belonging to one (cell type, gene) pair are independently permuted (cell barcodes are shuffled independently per unspliced/spliced). Repeating over all pairs, this results in a permuted data matrix. We then estimate the model fit of the unspliced and spliced abundance for permuted data matrix (the posterior predictive mean, Supplementary Methods). Note that because veloVI can handle held-out data, computing the model fit of permuted data does not require any additional training. Finally, for each (cell type and gene) pair we compute $\mu_p$ and $\mu_0$, which denote the mean absolute error between the model fit abundances and the observed abundances (spliced and unspliced errors added together) for the permuted and original data matrices, respectively.

To quantify the extent to which the mean absolute errors of the two samples are not equal, we define the permutation effect as the $t$-test statistic

$$T = \frac{\mu_p - \mu_0}{\sqrt{2\frac{S^2}{n}}},$$

with number of cells $n$ and pooled variance $S^2$ of the absolute errors. To limit the effect of dataset size, we consider the maximum sample size of $n = 200$ observations. The permutation score is aggregated on a gene level by considering the maximum test statistic across cell types. This aggregation allows comparing the permutation score across different datasets.

### Reporting summary

Further information on research design is available in the Nature Portfolio Reporting Summary linked to this article.

## Data availability

The processed pancreas data, including spliced and unspliced count abundances, can be downloaded from scVelo's GitHub (https://github.com/theislab/scvelo_notebooks/raw/master/data/Pancreas/endocrinogenesis_day15.h5ad). The forebrain and dentate gyrus datasets can be downloaded from the Kharchenko laboratory at Harvard (forebrain, http://pklab.med.harvard.edu/velocyto/DG1/10X43_1.loom and dentate gyrus, http://pklab.med.harvard.edu/velocyto/hgForebrainGlut/hgForebrainGlut.loom). The Friedrich Miescher Institute for Biomedical Research (https://www.fmi.ch/groups/gbioinfo/RNAVeloQuant/RNAVeloQuant.html) provides the processed data of the dentate gyrus, mouse brain, pancreas, prefrontal cortex and spermatogenesis. The mouse retina and PBMC data are available for download via figshare (https://figshare.com/projects/veloVI_datasets/145476).

## Code availability

veloVI is implemented in a standalone package at https://github.com/YosefLab/velovi, which has also been deposited via Zenodo (https://doi.org/10.5281/zenodo.7897641) (ref. 52). Code to reproduce the results in the manuscript can be found at https://github.com/YosefLab/velovi_reproducibility, as well as deposited via Zenodo (https://doi.org/10.5281/zenodo.7931042) (ref. 53).

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

## Acknowledgements

We thank R. Lopez and M. Jones for feedback on the concepts and benchmarking of veloVI. We acknowledge members of the Streets, Theis and Yosef laboratories for general feedback. A.S. is a Chan Zuckerberg Biohub investigator. A.G. and N.Y. were supported by the Chan Zuckerberg Initiative Essential Open Source Software Cycle 4 grant (EOSS4-0000000121) for scvi-tools. M.L. acknowledges financial support from the Joachim Herz Stiftung via Add-on Fellowships for Interdisciplinary Life Science. A.S. is a Chan Zuckerberg Biohub investigator and is supported by the National Institute of General Medical Sciences of the National Institutes of Health under award number R35GM124916. F.J.T. acknowledges support by the BMBF (grant nos. 01IS18036B and 01IS18053A) and by the Helmholtz Associations Initiative and Networking Fund through Helmholtz AI (grant no. ZT-I-PF-5-01).

## Author contributions

A.G. and P.W. contributed equally. A.G., P.W. and M.L. conceptualized the study. A.G. conceptualized the statistical model with contributions from M.L. and P.W. A.G. designed and implemented veloVI with contributions from P.W., J.H. and M.L. P.W. designed and implemented modeling extensions. D.K. designed and implemented model uncertainty analyses with contributions from A.G., P.W. and M.L. A.G., P.W. and J.H. designed and implemented analysis methods with contributions from M.L. A.S., F.J.T. and N.Y. supervised the work. A.G., P.W., M.L., F.J.T. and N.Y. wrote the manuscript.

## Competing interests

M.L. consults for Santa Ana Bio, is a part-time employee at Relation Therapeutics and owns interests in Relation Therapeutics. F.J.T. consults for Immunai, Singularity Bio, CytoReason and Omniscope and has ownership interest in Dermagnostix and Cellarity. N.Y. is an advisor and/or has equity in Cellarity, Celsius Therapeutics and Rheos Medicine. The remaining authors declare no competing interests.

## Additional information

**Extended data** is available for this paper at https://doi.org/10.1038/s41592-023-01994-w.

**Correspondence and requests for materials** should be addressed to Fabian J. Theis or Nir Yosef.

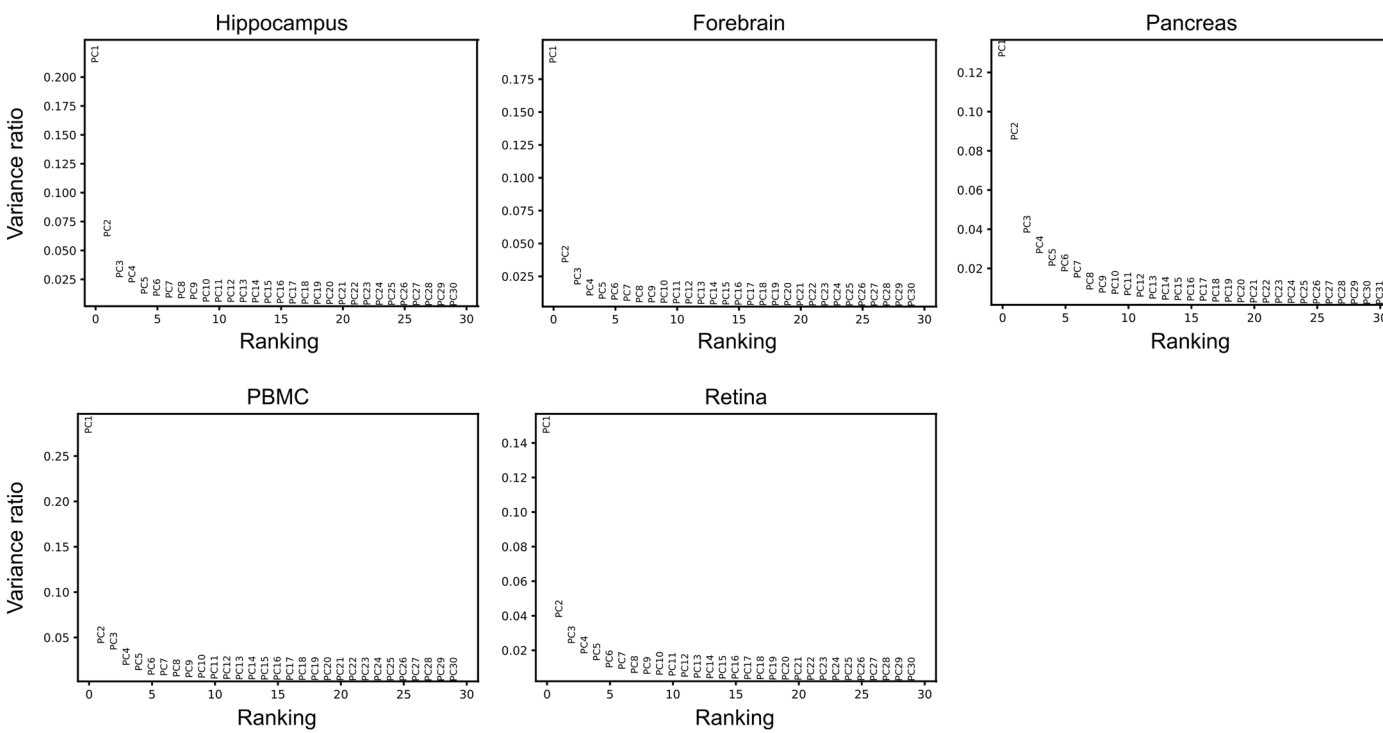

**Extended Data Fig. 1 | Low-rank structure of latent time.** PCA variance ratio of gene-cell specific latent time as inferred by the EM model.

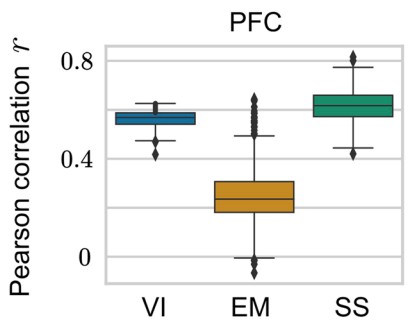
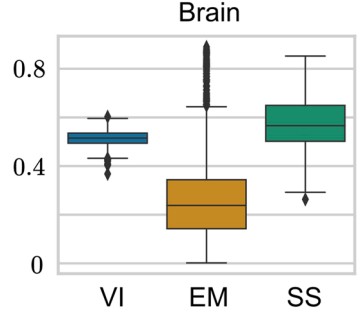
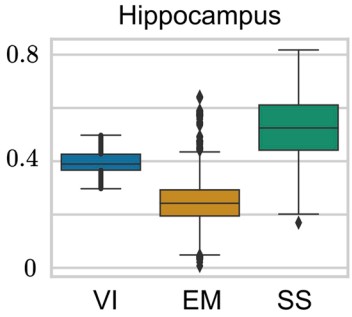

**Extended Data Fig. 2 | Preprocessing stability of inference methods.**
Correlation of velocities derived from pairs of quantification algorithms and from velocities estimating using one of veloVI (VI), the EM (EM), and steady-state model (SS) on datasets of prefrontal cortex (PFC) (left, N=78 pairs of quantification methods), 21-22 months old mouse brains (middle, N=78 pairs of quantification methods), and hippocampus (right, N=55 pairs of quantification methods). Unspliced and spliced counts are quantified with different algorithms[46–51,54]. Velocities are estimated by veloVI (VI, blue), the EM model (EM, orange), and the steady-state model (SS, green). Box plots indicate the median (center line), interquartile range (hinges), and whiskers at 1.5x interquartile range.

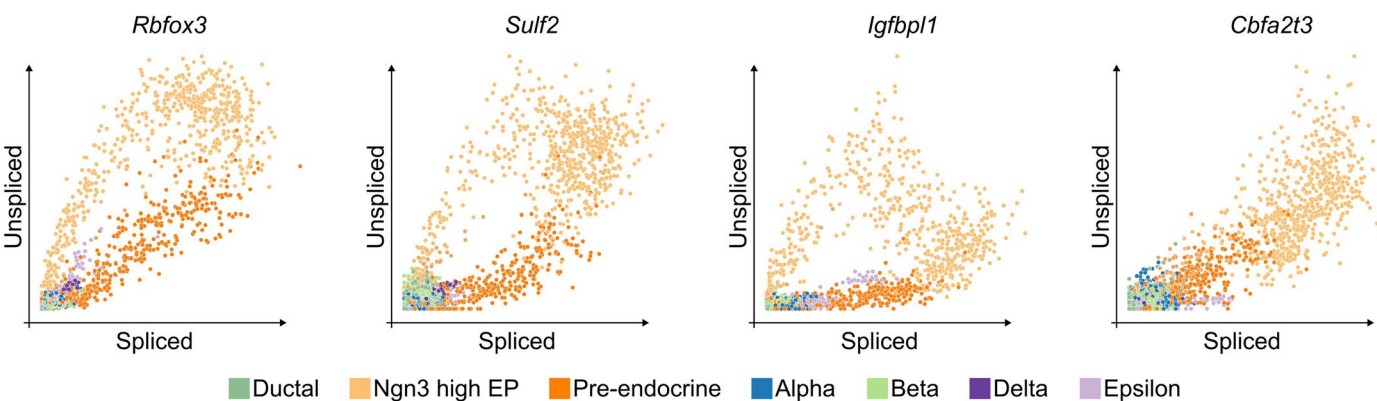

**Extended Data Fig. 3 | Phase portraits in pancreas endocrinogenesis.** Phase portraits of *Rbfox3*, *Sulf2*, *Igfbpl1*, and *Cbfa2t3*. Each cell is colored by its cell type.

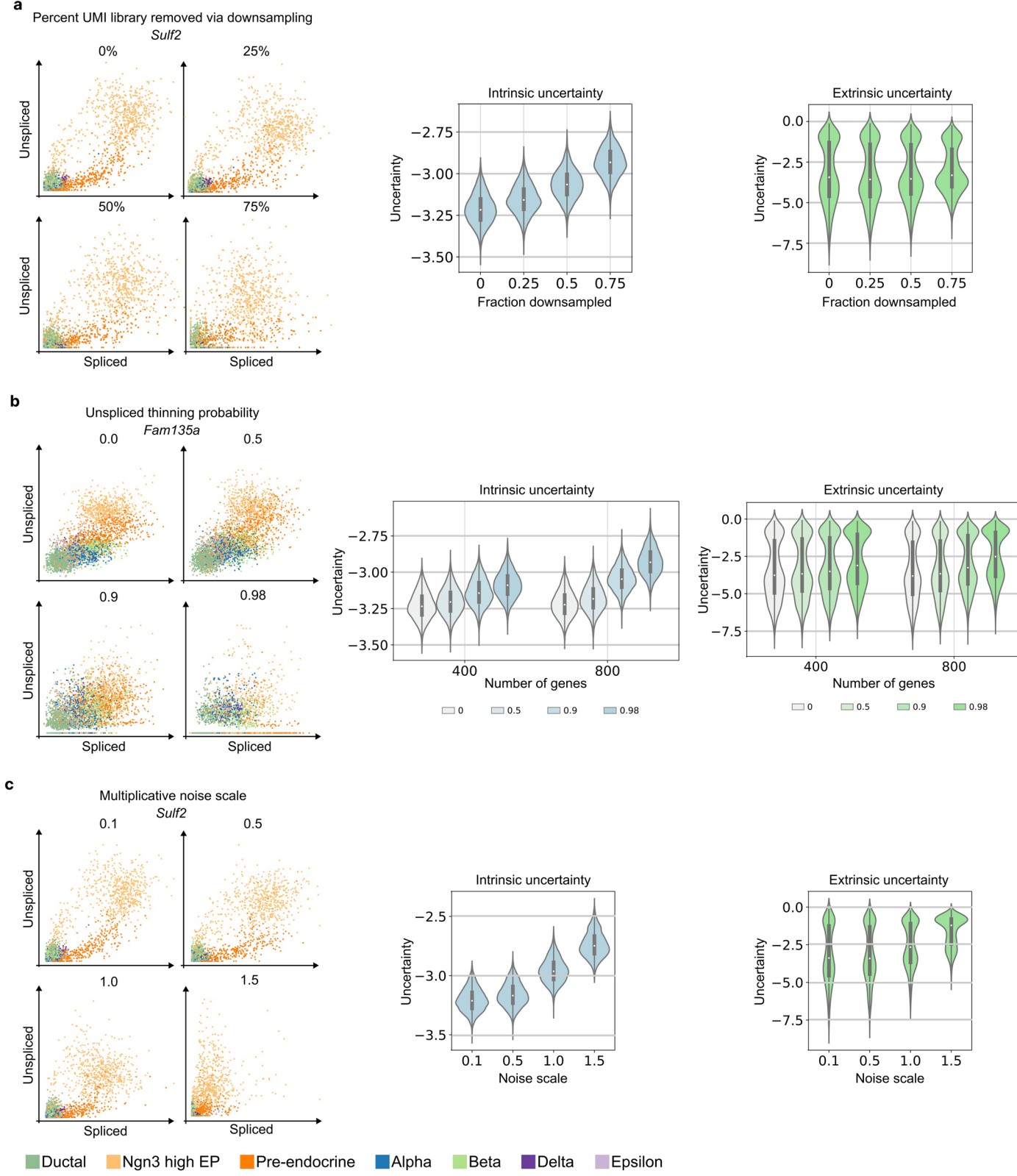

**Extended Data Fig. 4 | Effect of data perturbation on uncertainty. a**. The effect of downsampling (0%, 25%, 50%, 75%) counts on phase portraits of *Sulf2* (left) colored by cell type, intrinsic uncertainty per cell (middle, N=3696 cells), and extrinsic uncertainty per cell (right, N=3696 cells). **b**. The effect of unobserved unspliced reads (dropout probability 0.0, 0.5, 0.9, 0.98) in 400 and 800 genes on phase portraits of *Fam135a* (left), intrinsic uncertainty per cell (middle, N=3696 cells), and extrinsic uncertainty per cell (right, N=3696 cells). **c**. The effect of multiplicative noise (scale 0.1, 0.5, 1.0, 1.5) on phase portraits of *Sulf2* (left), intrinsic uncertainty per cell (middle, N=3696 cells), and extrinsic uncertainty per cell (right, N=3696 cells). Box plots indicate the median (center line), and interquartile range (hinges).

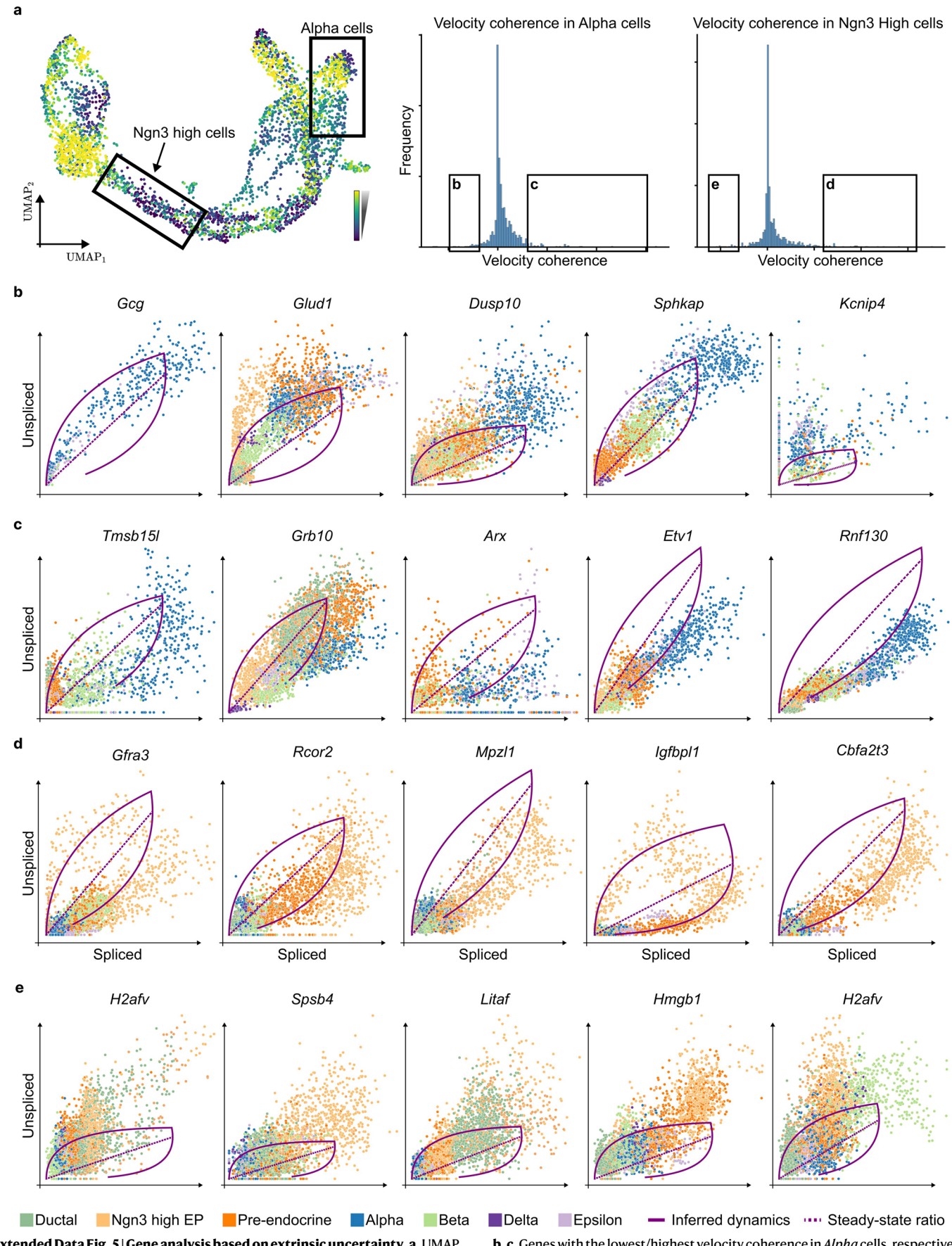

**Extended Data Fig. 5 | Gene analysis based on extrinsic uncertainty. a.** UMAP embedding of the Pancreas dataset colored by extrinsic uncertainty (left); The velocity coherence score across all genes for *Alpha* and Ngn3-high cells (right). **b**, **c**. Genes with the lowest/highest velocity coherence in *Alpha* cells, respectively. **c**, **d**. Genes with the lowest/highest velocity coherence in *Ngn3-high* cells, respectively. **e**, Genes fit with incorrect dynamics in Ngn3-high cells.

**a**

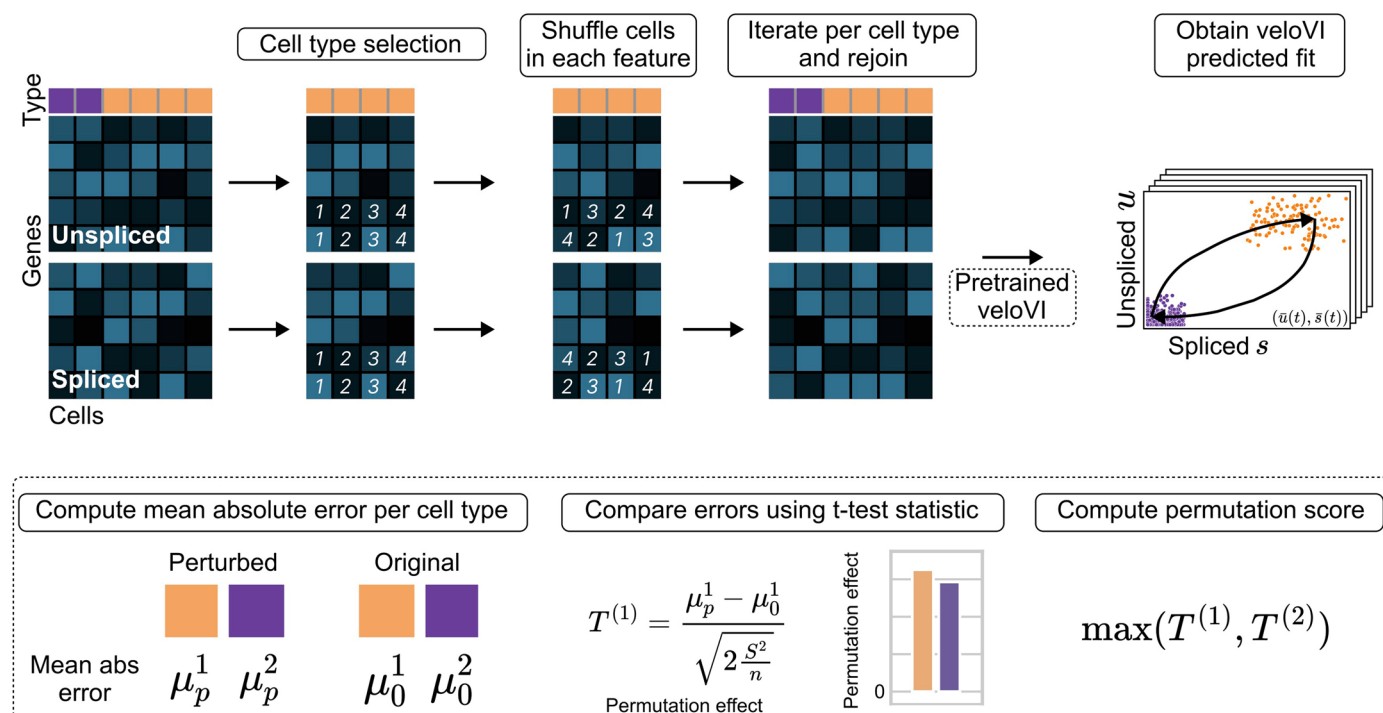

**Extended Data Fig. 6 | Overview of permutation score construction.**
**a**. First, the cells of one cell type are selected. These are shuffled independently for each genes (and independently in each of unspliced and spliced matrices). This is repeated for each cell type and the data are concatenated. This new permuted dataset is fed into a pre-trained veloVI model (trained on the same original dataset). The fit of unspliced and spliced abundance is obtained for each new perturbed cell. Following this, for each gene, the mean absolute error (spliced and unspliced) is computed per cell type. The original and perturbed mean absolute errors are compared with the T-test statistic. This provides a permutation effect statistic for each gene and each cell type. To obtain the permutation score, a scalar score for each gene, we take the maximum permutation effect statistic across cell types.

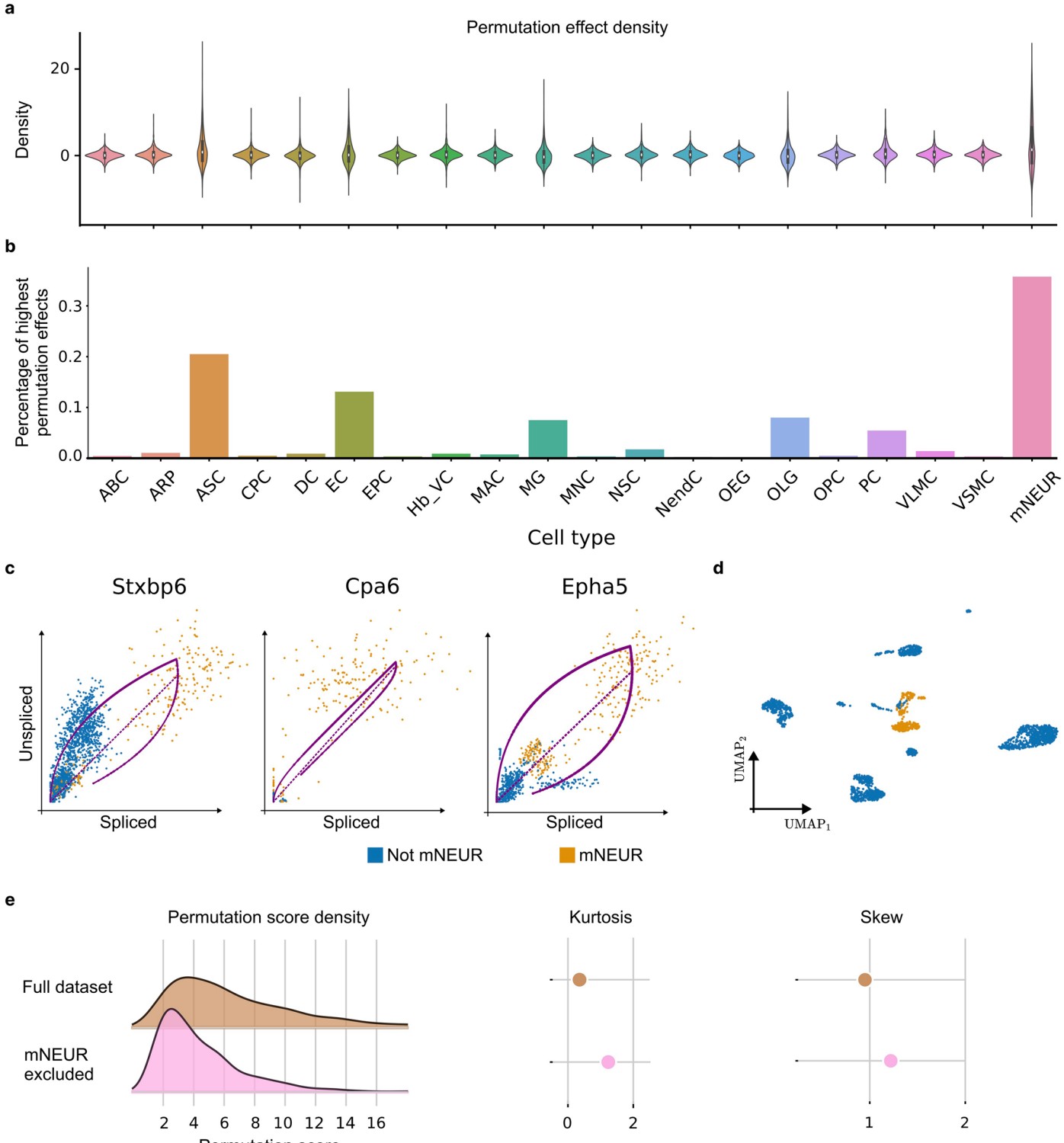

**Extended Data Fig. 7 | Permutation score analysis of old mouse brain.**
**a**. Density of permutation score per cell type: arachnoid barrier cells (ABC), astrocyte-restricted precursors (ARP), astrocytes (ASC), choroid plexus epithelial cells (CPC), dendritic cells (DC), endothelial cells (EC), ependymocytes (EPC), hemoglobin-expressing vascular cells (Hb-VC), macrophages (MAC), microglia (MG), monocytes (MNC), neural stem cells (NSC), neuroendocrine cells (NendC), olfactory ensheathing glia (OEG), oligodendrocytes (OLG), oligodendrocyte precursor cells (OPC), pericytes (PC), vascular and leptomeningeal cells (VLMC), vascular smooth muscle cells (VSMC), mature neurons (mNEUR) (N=2000 genes each). **b**. Percentage of cell types scoring assigned the highest permutation socre for a given gene. **c**. Genes assigned the highest permutation score. **d**, UMAP embedding of dataset colored by whether cells are mature neurons (mNEUR). **e**, Permutation score densities (left), and their kurtosis and skew when using the full dataset (brown) compared to excluding mature neurons.

## Reporting Summary

## Statistics

For all statistical analyses, confirm that the following items are present in the figure legend, table legend, main text, or Methods section.

| n/a | Confirmed | |
|---|---|---|
| ☐ | ☒ | The exact sample size (*n*) for each experimental group/condition, given as a discrete number and unit of measurement |
| ☐ | ☒ | A statement on whether measurements were taken from distinct samples or whether the same sample was measured repeatedly |
| ☐ | ☒ | The statistical test(s) used AND whether they are one- or two-sided<br>*Only common tests should be described solely by name; describe more complex techniques in the Methods section.* |
| ☐ | ☒ | A description of all covariates tested |
| ☐ | ☒ | A description of any assumptions or corrections, such as tests of normality and adjustment for multiple comparisons |
| ☐ | ☒ | A full description of the statistical parameters including central tendency (e.g. means) or other basic estimates (e.g. regression coefficient) AND variation (e.g. standard deviation) or associated estimates of uncertainty (e.g. confidence intervals) |
| ☐ | ☒ | For null hypothesis testing, the test statistic (e.g. *F*, *t*, *r*) with confidence intervals, effect sizes, degrees of freedom and *P* value noted<br>*Give P values as exact values whenever suitable.* |
| ☐ | ☒ | For Bayesian analysis, information on the choice of priors and Markov chain Monte Carlo settings |
| ☒ | ☐ | For hierarchical and complex designs, identification of the appropriate level for tests and full reporting of outcomes |
| ☐ | ☒ | Estimates of effect sizes (e.g. Cohen's *d*, Pearson's *r*), indicating how they were calculated |

*Our web collection on statistics for biologists contains articles on many of the points above.*

## Software and code

Policy information about availability of computer code

| Data collection | No software was used. |
|---|---|
| Data analysis | anndata 0.8.0<br>scanpy 1.9.1<br>h5py 3.7.0<br>igraph 0.9.11<br>joblib 1.1.0<br>leidenalg 0.8.10<br>llvmlite 0.38.1<br>loompy 3.0.7<br>louvain 0.7.1<br>matplotlib 3.5.2<br>numba 0.55.2<br>numpy 1.22.4<br>pandas 1.3.5<br>scipy 1.8.1<br>scvi-tools 0.16.4<br>sklearn 1.1.1<br>scvelo 0.2.5.dev71+g85295fc<br>velovi 0.1.0 |

veloVI is implemented in a standalone package at https://github.com/YosefLab/velovi, which has also been deposited via Zenodo (https://doi.org/10.5281/zenodo.7897641). Code to reproduce the results in the manuscript can be found at: https://github.com/YosefLab/velovi_reproducibility, as well as deposited via Zenodo (https://doi.org/10.5281/zenodo. 7931042)

For manuscripts utilizing custom algorithms or software that are central to the research but not yet described in published literature, software must be made available to editors and reviewers. We strongly encourage code deposition in a community repository (e.g. GitHub). See the Nature Portfolio guidelines for submitting code & software for further information.

## Data

Policy information about availability of data

All manuscripts must include a data availability statement. This statement should provide the following information, where applicable:
- Accession codes, unique identifiers, or web links for publicly available datasets
- A description of any restrictions on data availability
- For clinical datasets or third party data, please ensure that the statement adheres to our policy

The processed Pancreas data, including spliced and unspliced count abundances, can be downloaded from scVelo's GitHub (https://github.com/theislab/scvelo_notebooks/raw/master/data/Pancreas/endocrinogenesis_day15.h5ad). The forebrain and dentate gyrus datasets can be downloaded from the Kharchenko lab at Harvard (forebrain: http://pklab.med.harvard.edu/velocyto/DG1/10X43_1.loom, dentate gyrus: http://pklab.med.harvard.edu/velocyto/hgForebrainGlut/hgForebrainGlut.loom). The Friedrich Miescher Institute for Biomedical Research (https://www.fmi.ch/groups/gbioinfo/RNAVeloQuant/RNAVeloQuant.html) provides the processed data of the dentate gyrus, mouse brain, pancreas, prefrontal cortex, and spermatogenesis. The mouse retina and PBMC data is available for download via figshare (https://figshare.com/projects/veloVI_datasets/145476).

## Human research participants

Policy information about studies involving human research participants and Sex and Gender in Research.

| | |
|---|---|
| Reporting on sex and gender | N/A |
| Population characteristics | N/A |
| Recruitment | N/A |
| Ethics oversight | N/A |

Note that full information on the approval of the study protocol must also be provided in the manuscript.

# Field-specific reporting

Please select the one below that is the best fit for your research. If you are not sure, read the appropriate sections before making your selection.

☒ Life sciences          ☐ Behavioural & social sciences          ☐ Ecological, evolutionary & environmental sciences

For a reference copy of the document with all sections, see nature.com/documents/nr-reporting-summary-flat.pdf

# Life sciences study design

All studies must disclose on these points even when the disclosure is negative.

| | |
|---|---|
| Sample size | No sample size calculation was performed. Sample sizes were determined based on preprocessed scRNA-seq data. |
| Data exclusions | No data were excluded from the analysis. |
| Replication | All of the findings reported in this study are reproducible based on code in the veloVI reproducibility GitHub repository (code availability). |
| Randomization | Randomization was not relevant to this study. |
| Blinding | Blinding was not relevant to this study as there were no case-control comparisons made. |

# Reporting for specific materials, systems and methods

We require information from authors about some types of materials, experimental systems and methods used in many studies. Here, indicate whether each material, system or method listed is relevant to your study. If you are not sure if a list item applies to your research, read the appropriate section before selecting a response.

## Materials & experimental systems

| n/a | Involved in the study |
|-----|----------------------|
| ☒ | Antibodies |
| ☒ | Eukaryotic cell lines |
| ☒ | Palaeontology and archaeology |
| ☒ | Animals and other organisms |
| ☒ | Clinical data |
| ☒ | Dual use research of concern |

## Methods

| n/a | Involved in the study |
|-----|----------------------|
| ☒ | ChIP-seq |
| ☒ | Flow cytometry |
| ☒ | MRI-based neuroimaging |

