## [Peer Review file · Nature Methods]

Peer Review Information

Manuscript Title: Deep generative modeling of transcriptional dynamics for RNA velocity analysis in single cells

Corresponding author name(s): Nir Yosef

Editorial Notes:

Reviewer Comments & Decisions:

Decision Letter, initial version:

11th Oct 2022

Dear Dr Yosef,

Your Brief Communication, "Deep generative modeling of transcriptional dynamics for RNA velocity analysis in single cells", has now been seen by 2 reviewers. As you will see from their comments below, although the reviewers find your work of considerable potential interest, they have raised a number of concerns. We are interested in the possibility of publishing your paper in Nature Methods, but would like to consider your response to these concerns before we reach a final decision on publication.

We therefore invite you to revise your manuscript to fully address all these concerns. We are committed to providing a fair and constructive peer-review process. Do not hesitate to contact us if there are specific requests from the reviewers that you believe are technically impossible or unlikely to yield a meaningful outcome.

- * include a point-by-point response to the reviewers and to any editorial suggestions
- * please underline/highlight any additions to the text or areas with other significant changes to facilitate review of the revised manuscript
- * address the points listed described below to conform to our open science requirements

* ensure it complies with our general format requirements as set out in our guide to authors at www.nature.com/naturemethods

* resubmit all the necessary files electronically by using the link below to access your home page

[REDACT]

We hope to receive your revised paper within eight weeks. We are very aware of the difficulties caused by the COVID-19 pandemic to the community. If you cannot send it within this time, please let us know. In this event, we will still be happy to reconsider your paper at a later date so long as nothing similar has been accepted for publication at Nature Methods or published elsewhere.

OPEN SCIENCE REQUIREMENTS

REPORTING SUMMARY AND EDITORIAL POLICY CHECKLISTS

DATA AVAILABILITY

All novel DNA and RNA sequencing data, protein sequences, genetic polymorphisms, linked genotype and phenotype data, gene expression data, macromolecular structures, and proteomics data must be deposited in a publicly accessible database, and accession codes and associated hyperlinks must be provided in the "Data Availability" section.

CODE AVAILABILITY

Please include a "Code Availability" subsection in the Online Methods which details how your custom code is made available. Only in rare cases (where code is not central to the main conclusions of the paper) is the statement "available upon request" allowed (and reasons should be specified).

For more information on our code sharing policy and requirements, please see: <https://www.nature.com/nature-research/editorial-policies/reporting-standards#availability-of-computer-code>

MATERIALS AVAILABILITY

ORCID

Nature Methods is committed to improving transparency in authorship. As part of our efforts in this direction, we are now requesting that all authors identified as 'corresponding author' on published papers create and link their Open Researcher and Contributor Identifier (ORCID) with their account on

the Manuscript Tracking System (MTS), prior to acceptance. This applies to primary research papers only. ORCID helps the scientific community achieve unambiguous attribution of all scholarly contributions. You can create and link your ORCID from the home page of the MTS by clicking on 'Modify my Springer Nature account'. For more information please visit www.springernature.com/orcid.

Sincerely,

Lin Tang, PhD
Senior Editor
Nature Methods

Reviewers' Comments:

Reviewer #1:

Remarks to the Author:

RNA velocity estimation emerges as an exciting approach for extracting (at least) partial dynamical information from scRNA-seq data, and some technical challenges remain. A number of approaches have been developed. In this work, Gayoso et al. reformulated the EM model in the formalism of variational autoencoder. A prominent feature of this formulation is that one can estimate the quality of the inferred RNA velocities, which is very useful in practice. Due to the dispersed expression levels of different genes and the intrinsic limitations of splicing-based RNA velocities, the estimated velocities of individual genes may have different qualities, which in practice have led to spurious trajectory inference such as direction from differentiated states to known progenitor states. I suggest that this estimation of posterior velocity uncertainty is the most notable contribution from VeloAI. While a number of other existing methods can provide some information on the velocity quality (esp. VeloVAE that uses similar theoretical framework), VeloAI has some unique features, as the authors argued in their supplementary note.

Here are a few points that I seek clarification:

- 1) VeloVI tries to estimate the parameters α , β , γ , which is similar to the EM model, and differs from the original RNA velocity method. The latter only needs to estimate γ and assume $\beta = 1$, so the velocities are only relative. Now with all the three parameters estimated for a given gene, one can get an absolute velocity. Another example of getting such absolute velocity is from data with metabolic labeling, which arguably provides more direct information about mRNA turnover dynamics (Qiu, et al., Cell, 185:690, 2022). Therefore, is there any issue on constraining all the parameters from the splicing/unsplicing data? I assume with VeloVI one can examine how well these parameters are constrained by the data, for which readers may be interested in knowing.
- 2) Given above, on page 3 of the supplemental note, the authors indicate that they assume that each gene is on the same time scale, with a maximum of time scale $t = 20$. Does it mean their estimation is somewhat similar to the original model with a constant β in the sense that the velocity should be treated as relative as well? How does this parameter $t = 20$ may or may not affect the results? Some explicit clarification will be helpful to the readers.
- 3) Continuing to point 1, while it is nice to have the three parameters specified, it comes at some

expense that the authors should discuss explicitly. Notice in the derivations all the parameters α , β , and γ are assumed to be constant (or at least to a specific group of cells). This is equivalent to assume that each gene can exist in only two expression states (completely on or off). However, biologically we know that gene transcription rates (and even the other two parameters) are under regulation, leading to the dose-response curves one typically talks about. That is, “ α ” is not a constant but a function of the levels of RNA polymerase and many other regulatory elements, (and generally speaking similar for β and γ). A main effort in systems biology is to get such relation. Actually, in Qiu et al. (Cell 2022), they showed how one can use such relation to reconstruct the quantitative gene regulation relations from RNA velocity data. Notice that in the original RNA velocity paper, this constant α assumption is avoided since $ds/dt = u - \gamma s$, and (with some assumption) all the regulation is contained in the measured u (as explicitly discussed in a recent review Xing, Phys Biol, 19:061001, 2022). With metabolic labeling, Qiu, et al. (Cell, 185:690, 2022) showed one can estimate cell-specific values of α and β (but still assume a cell-type-specific γ value). The authors should discuss on the potential limitations and caveats with their model, an issue also raised up in a recent paper (Gorin et al, PLoS Comput Biol 18(9): e1010492). I understand that the authors mentioned that their formulation can be generated to a time-dependent α with certain time-dependence function form, but we are talking about a more generic situation.

Some minor points:

- 4) It is straightforward to generalize the current formalism to analyze data with metabolic labeling. The authors may point out this.
- 5) Equation 14 of the supplemental note: Although it is often assumed, it might be helpful to add a sentence of justification of using Dirichlet distribution in the context.
- 6) Similarly, Eqn 20/21: some justification of the normal distributions will help a general reader.
- 7) Page 4 of the supplemental note, “For all experiments in this manuscript we use $c_k = 1$ except for the repression steady state in which $c_4 = 0.1$ ”: How these specific values be chosen, and how varying these values may or may not affect the results?
- 8) Eqn 22: it might be helpful to remind a reader that the sum of n is over all the cells and the sum of g is over all the (four) cell states.
- 9) Page 5: Are the minibatches of 256 cells randomly selected? Are the values of u^* and s^* what used in the EM model

Jianhua Xing

Reviewer #2:

Remarks to the Author:

Summary

The authors introduce veloVI, a deep learning model to infer RNA velocity parameters from single cell RNA-seq data using an adaptation of the scVI VAE framework. In brief, the authors provide spliced and unspliced count matrices as input, infer a latent set of RNA velocity and cell state parameters, and reconstruct spliced and unspliced count matrices from the latent state along with point-estimates of RNA velocity for each cell and gene sampled from the latent posterior. In comparisons to previous RNA velocity models, the authors report that veloVI yields velocity estimates that more accurately reflect simulations, are more robust to pre-processing choices, and enable estimates of RNA velocity uncertainty that aid interpretation.

The manuscript is well-written, the model is well reasoned, and the method represents a step forward for the RNA velocity inference field by introducing one of the first methods for quantifying velocity uncertainty in a robust manner. As presented, the model is compelling, but additional experiments are

necessary to validate the full claims of the manuscript.

Major Points

1. Comparison of veloVI and EM accuracy

In Fig. 1D the velocity inference results from veloVI and the EM algorithm are not particularly well correlated ($r^2 \approx 0.4$ by eye). Both procedures can't be correct in this case. Simulation results suggest that veloVI is marginally better than the EM algorithm, but **both** algorithms are highly correlated with simulated ground truth in that context (Fig. S2B). This suggests the simulations are not representative of any of the real datasets, and it's hard to know from these results alone if veloVI is actually more or less accurate than the EM approach.

The authors should include (1) experiments with more realistic simulations that reflect the disparity between veloVI and EM seen in all real datasets shown here, or ideally, (2) experiments in real data that have metabolic labels, allowing for an orthogonal, molecular method of inferring velocity that may better represent real biology. See scSLAM-seq (Erhard 2019), sci-Fate (Cao 2020), NASC-seq (Hendriks 2019), scNT-seq (Qiu 2020) as example datasets.

2. Velocity consistency comparisons

Fig. 1D (center panel) shows that veloVI has a higher "velocity consistency" across all datasets. Is this necessarily a desirable feature? In the simplest case, a kNN smoothing post-processing step on inferred velocities would increase the consistency for any inference procedure. This only seems beneficial if it's abundantly clear that veloVI's estimates are *also* more likely to be accurate (See Major Point 1). The authors should comment on why higher consistency is always desirable and ideally demonstrate a case where known biology is better reflected using veloVI's more consistent estimates than the prior EM algorithm's estimates.

3. Validating velocity uncertainty estimates

Estimating the uncertainty in velocity vectors is a major contribution, as prior methods in the literature for intrinsic uncertainty are expensive. However, it's difficult to determine from the current results alone if the uncertainty estimates provided by veloVI are well-calibrated. Many deep learning models are known to suffer from poor calibration absent specific techniques to improve uncertainty estimates, so it's not clear from first principles that veloVI uncertainty estimates are correct. The authors should include experiments comparing the estimates of uncertainty from veloVI to other metrics of uncertainty derived by e.g. simulating data with different uncertainty on velocity parameters, Monte Carlo sampling of reads within each cell as an empirical estimate of uncertainty, or injecting random noise into data to show that veloVI estimates reflect increased uncertainty.

4. Permutation score

The permutation score is very clever, and the results in Fig. 2d are compelling evidence it captures some real measure of how "dynamical" a cell system is. However, it's unclear how users should apply this score. Is it a pre-filtering metric for individual input genes? Is it a way to exclude whole clusters from analysis? Whole datasets? The authors should provide a clearer description of how and when this score is useful for downstream users of veloVI, and include experiments to demonstrate that utility. As it stands, the score is more of a curiosity than a useful tool.

Minor Points

1. Fig. 1A is beautiful, but difficult to parse with the current legend. Readers would benefit from definitions of π and k_{ng} in the legend or inset alongside the panels.
2. The authors should consider including the kNN smoothing + scaling preprocessing steps in Fig. 1A. This is impactful for interpreting the results.
3. The explanation of the permutation score in Fig. 2c is highly unclear. Fig. 2c was required for me to understand the procedure, but readers with less RNA velocity experience might not grok it from the plot. I'd suggest rephrasing to explain the intuition in the main text, and adding more clear explanation in the main methods for which values are being permuted. Maybe provide a sample set of vectors running through the procedure in Supplemental Methods.

Help your paper reach publication quicker

Springer Nature Author Services can help you improve your manuscript through services including English language editing, developmental comments, manuscript formatting, figure preparation, translation, and more.

[Get started—and save 15%](https://authorservices.springernature.com/go/sn/?utm_source=EJP&utm_medium=Revision+Email&utm_campaign=SNAS+Referrals+2022&utm_id=ref2022)

Please note that using these tools, or any other service, is not a requirement for publication, nor does it imply or guarantee that editors will accept the article, or even select it for peer review.

Author Rebuttal to Initial comments

Point-by-point response to the reviewers' comments

Deep generative modeling of transcriptional dynamics for RNA velocity analysis in single cells

Adam Gayoso^{1,*}, Philipp Weiler^{2,3,*}, Mohammad Lotfollahi², Dominik Klein², Justin Hong¹, Aaron Streets^{1,4,5}, Fabian J. Theis^{2,3,6,#}, Nir Yosef^{1,7,#}

* equal contribution

correspondence to fabian.theis@helmholtz-muenchen.de, niryosef@berkeley.edu

¹Center for Computational Biology, University of California, Berkeley, Berkeley, CA, USA

²Institute of Computational Biology, Helmholtz Center Munich, Munich, Germany

³Department of Mathematics, Technical University of Munich, Munich, Germany

⁴Department of Bioengineering, University of California, Berkeley, Berkeley, CA, USA

⁵Chan Zuckerberg Biohub, San Francisco, CA, USA

⁶TUM School of Life Sciences Weihenstephan, Technical University of Munich, Munich, Germany

⁷Department of Electrical Engineering and Computer Sciences, University of California, Berkeley, Berkeley, CA, USA

In the following, we present our response to the reviewers comments. We give **comments (black)**, **point-by-point answers (green)** to the questions and in parts **copy parts of the text or specific panels (blue)**, which directly correspond to comments or reference to them.

Editor comments:

Your Brief Communication, "Deep generative modeling of transcriptional dynamics for RNA velocity analysis in single cells", has now been seen by 2 reviewers. As you will see from their comments below, although the reviewers find your work of considerable potential interest, they have raised a number of concerns. We are interested in the possibility of publishing your paper in Nature Methods, but would like to consider your response to these concerns before we reach a final decision on publication.

We therefore invite you to revise your manuscript to fully address all these concerns. We are committed to providing a fair and constructive peer-review process. Do not hesitate to contact us if there are

specific requests from the reviewers that you believe are technically impossible or unlikely to yield a meaningful outcome.

We thank the reviewers for their insightful and thorough comments. Based on the reviewers' feedback, we have added additional analyses and improved our exposition of the capabilities and limitations of veloVI. Our updates of main and supplemental figures include new results that further validate the utility of veloVI over previous approaches. We have also added one new supplementary note and five new supplementary figures that provide a deeper dive into our modeling decisions, implementation, and implications for analysis. The main revisions were:

- 1. Enhanced benchmarking validation of veloVI performance relative to the scVelo EM model.** We included new benchmarks based on simulations and real data that demonstrate that veloVI learns improved velocities and latent time estimations compared to the EM model.
- 2. Improved clarity of modeling choices, limitations, and presentation.** We improved the clarity of the parts dealing with these important subjects in the main text. For better accessibility, we also wrote a new supplementary note (Supplementary Note 4) that discusses limitations in learning absolute RNA velocity values from conventional scRNA-seq data, modeling choices with respect to kinetic parameters, future opportunities for improving the veloVI formulation, as well as extensions to metabolic labeling data. Furthermore, we improved the clarity of presentation of the permutation score, which we use to scrutinize the velocity model fit. Finally, we include an overview of the veloVI architecture as Supplementary Figure 17.
- 3. Benchmarking veloVI's uncertainty quantification.** We included new simulations based on perturbing real data to show that veloVI's uncertainty quantification is associated with random noise, as well as bias in how unspliced molecules are counted/detected.

We believe that our revised manuscript more rigorously characterizes the performance of veloVI and more clearly communicates the impact of our study. Please find our point-by-point response to the reviewers' comments below, with references to the revisions in the main text (new additions in blue), figures and supplemental material.

Reviewer #1 (Remarks to the Author):

RNA velocity estimation emerges as an exciting approach for extracting (at least) partial dynamical information from scRNA-seq data, and some technical challenges remain. A number of approaches have been developed. In this work, Gayoso et al. reformulated the EM model in the formalism of variational autoencoder. A prominent feature of this formulation is that one can estimate the quality of the inferred RNA velocities, which is very useful in practice. Due to the dispersed expression levels of different genes and the intrinsic limitations of splicing-based RNA velocities, the estimated velocities of individual genes may have different qualities, which in practice have led to spurious trajectory inference such as direction from differentiated states to known progenitor states. I suggest that this estimation of posterior velocity uncertainty is the most notable contribution from VeloAI. While a number of other existing methods can provide some information on the velocity quality (esp. VeloVAE that uses similar theoretical framework), VeloAI has some unique features, as the authors argued in their supplementary note.

We thank the reviewer for the positive comments about the veloVI model and manuscript. We agree with the reviewer that leveraging the uncertainty from the veloVI will become an important aspect of analyzing RNA velocity derived from conventional scRNA-seq data.

With respect to the velocity uncertainty, we have included further validation of the uncertainty (as a response to reviewer 2 comment 3). We find that the uncertainty can be explained in part by biased detection of unspliced RNA, as well as general sampling noise in the data. We include point-by-point responses to the rest of this reviewer's comments below.

Major comments:

1.1 VeloVI tries to estimate the parameters alpha, beta, gamma, which is similar to the EM model, and differs from the original RNA velocity method. The latter only needs to estimate gamma and assume beta = 1, so the velocities are only relative. Now with all the three parameters estimated for a given gene, one can get an absolute velocity. Another example of getting such absolute velocity is from data with metabolic labeling, which arguably provides more direct information about mRNA turnover

dynamics (Qiu, et al., Cell, 185:690, 2022). Therefore, is there any issue on constraining all the parameters from the splicing/unsplicing data? I assume with veloVI one can examine how well these parameters are constrained by the data, for which readers may be interested in knowing.

We thank the reviewer for this comment. Indeed, veloVI, like the EM model that predates it, estimates the splicing rate constant, the degradation rate constant, as well as the transcription rate.

Despite estimating all three parameters, we emphasize that the velocities estimated by the EM and VI models are not absolute. Briefly, there is a scaling non-identifiability in the model, where the rates and the assumed max time can be scaled (and respectively inversely scaled) by a constant without changing the overall model fit. We show this mathematically in Supplementary Note 4 (see point 1.3 for overview), and relate it to the original RNA velocity method, which is relative for a different reason ($\beta=1$ as mentioned by the reviewer). We also added text to the discussion to emphasize this point:

Furthermore, while veloVI's estimated velocities are relative to a given maximum time of the process (similar as for the *EM model*), they are no longer relative with respect to the splicing rate as in the *steady-state model*. In future iterations, we anticipate including prior information from metabolic labeling data to estimate absolute velocities. We discuss these challenges as well as other considerations and future opportunities in Supplementary Note 4.

With respect to the comment about constraining parameters, in principle the veloVI framework can be used for model criticism, i.e., modifying and experimenting with changes to the dynamical model, including constraining rate parameters across genes. As for how well the current parameterization is constrained by the data, we have included the distribution of the rates/rate constants as new Supplementary Figure 16.

1.2 Given above, on page 3 of the supplemental note, the authors indicate that they assume that each gene is on the same time scale, with a maximum of time scale $t = 20$). Does it mean their estimation is somewhat similar to the original model with a constant β in the sense that the velocity should be treated as relative as well? How does this parameter $t = 20$ may or may not affect the results? Some explicit clarification will be helpful to the readers.

Following the previous response, the reviewer is correct to assume that the velocities are relative. However, the source of relativity is different in veloVI compared to the original RNA velocity formulation, as described in the response to 1.1.

The max time parameter only affects the magnitude of the velocities, but not the direction. We outline and discuss these important subtleties as part of a new Supplementary Note 4 (see next point).

1.3 Continuing to point 1, while it is nice to have the three parameters specified, it comes at some expense that the authors should discuss explicitly. Notice in the derivations all the parameters alpha, beta, and gamma are assumed to be constant (or at least to a specific group of cells). This is equivalent to assume that each gene can exist in only two expression states (completely on or off). However, biologically we know that gene transcription rates (and even the other two parameters) are under regulation, leading to the dose-response curves one typically talks about. That is, “alpha” is not a constant but a function of the levels of RNA polymerase and many other regulatory elements, (and generally speaking similar for beta and gamma). A main effort in systems biology is to get such relation. Actually, in Qiu et al. (Cell 2022), they showed how one can use such relation to reconstruct the quantitative gene regulation relations from RNA velocity data. Notice that in the original RNA velocity paper, this constant alpha assumption is avoided since $ds/dt = u - \gamma s$, and (with some assumption) all the regulation is contained in the measured u (as explicitly discussed in a recent review Xing, Phys Biol, 19:061001, 2022). With metabolic labeling, Qiu, et al. (Cell, 185:690, 2022) showed one can estimate cell-specific values of alpha and beta (but still assume a cell-type-specific gamma value). The authors should discuss on the potential limitations and caveats with their model, an issue also raised up in a recent paper (Gorin et al, PLoS Comput Biol 18(9): e1010492). I understand that the authors mentioned that their formulation can be generated to a time-dependent alpha with certain time-dependence function form, but we are talking about a more generic situation.

We thank the reviewer for this comment; we also feel that analysis with RNA velocity has the potential to reveal gene regulatory patterns and can be a valuable tool for the much larger effort of learning the “gene regulatory network”.

To address the reviewer’s concern, we have included a new supplementary note in the manuscript (Supplementary Note 4). *Note that in the first submission, Supplementary Note 4 described time-dependent transcription rates. This content was moved into the Supplementary Methods under the*

section “Time-dependent transcription rate”. In this new Supplementary Note 4 we discuss the following points:

1. Inference of absolute RNA velocity
 - a. Here we discuss that despite estimating all three parameters, the velocity estimates with veloVI (and the EM model for that matter) are not absolute due to the unknown experimental time, and provide a mathematical justification.
 - b. We emphasize that this is an issue with conventional scRNA-seq technologies that provide snapshot data of unspliced and spliced counts.
 - c. We comment on the differences between the relativity of veloVI velocities, and the velocity originally described in La Manno, et al., which are relative to an unestimated splicing rate.
2. Constraints on formulation of rates and rate constants
 - a. We highlight that the assumption of constant rates (including transcription rate) is sufficient to identify the steady-state, if observed, in the upper right part of the phase portrait.
 - b. We outline that while considering non-constant rates is possible in theory, it requires adaptations to the current veloVI model. In the case of the time-dependent transcription rates in this manuscript (Supplementary Figure 15), we chose a functional form that allowed obtaining a closed-form ODE solution. More complex functional forms, like neural nets, will require differentiable ODE solvers to estimate splicing kinetic parameters in the veloVI framework.
 - c. Cell or cluster specific rates add another layer of complexity to the formulation in that cell/cluster specific initial conditions become necessary, and these are often unobserved or lack enough prior information to be guessed.
3. Model likelihood
 - a. We recapitulate that veloVI’s likelihood is based on independent mixtures of Gaussians and outline possible modifications and their advantages.
 - b. We explain why the assumption of piecewise constant transcription during induction and repression yields a non-differentiable likelihood (if not considering a mixture model) and how this issue could be circumvented in future work.
 - c. We outline how count-based likelihoods could eradicate the need for data smoothing prior to training veloVI.
4. Exploration with velocity vector fields (relating to Dynamo’s idea)

- a. We point out that veloVI can naturally perform inference on unobserved data without any additional optimization steps, and that future work can compare vector fields with veloVI to the procedure used in the Dynamo manuscript.
 - b. We stress that while working with unobserved data is possible in theory, the current model formulation will need to be extended to remove batch effects, for example.
5. Extensions for metabolic labeling data
- a. We reiterate that veloVI is applied to conventional scRNA-seq data, give a brief overview on metabolic labeling data, and how it may be useful in the context of RNA velocity.
 - b. We explain that the proposed veloVI framework is extensible to use metabolically labeled transcripts, and outline how this adapted model would compare to Dynamo.

Minor comments:

1.4 It is straightforward to generalize the current formalism to analyze data with metabolic labeling. The authors may point out this.

We have included this point in the discussion, with more information in the new Supplementary Note 4 as described in our response to point 1.3.

1.5 Equation 14 of the supplemental note: Although it is often assumed, it might be helpful to add a sentence of justification of using Dirichlet distribution in the context.

We have included the following new text in relation to the Dirichlet distribution:

Here π_{ng} is sampled from a Dirichlet distribution, which has the support of the probability simplex. In other words, the Dirichlet provides a distribution over discrete probability distributions.

1.6 Similarly, Eqn 20/21: some justification of the normal distributions will help a general reader.

We have included the following new text in relation to the normal distribution:

By using the normal distribution, we assume that the smoothed expression (which represents an average of random variables) has a sampling distribution centered on some mean value, and that this sampling distribution is approximately normal.

1.7 Page 4 of the supplemental note, “For all experiments in this manuscript we use $c_k = 1$ except for the repression steady state in which $c_4 = 0.1$ ”: How these specific values be chosen, and how varying these values may or may not affect the results?

We have included the following text to describe this choice:

This hyperparameter choice forces the variance of abundance in the repression steady state to be less than that of other transcriptional states, which reflects the notion that the repression steady state corresponds to zero transcriptional activity. Despite the assumption of zero transcriptional activity, the normal distribution captures noise that arises during the experimental process (ambient transcripts) as well as during preprocessing (e.g., knn smoothing).

1.8 Eqn 22: it might be helpful to remind a reader that the sum of n is over all the cells and the sum of g is over all the (four) cell states.

We have added one sentence to clarify this point:

Here z factorizes over all N cells and π_{ng} over all N cells and G genes.

1.9 Page 5: Are the minibatches of 256 cells randomly selected? Are the values of u^* and s^* what used in the EM model

Indeed, the 256 cells are randomly selected and we clarified this in the text:

In particular, we use randomly sampled minibatches of 256 cells for inference.

The values u^* and s^* are the analogous values to the EM model fit values. The u^* and s^* for the EM model can be described as a posterior predictive distribution mean, however the “posterior” distributions here would be Dirac delta distributions over the state assignment and time of a cell/gene. We have clarified this point in the text:

We compare these fitted abundance values from veloVI to the analog of the EM model, which itself can be interpreted as a posterior predictive mean. Considering just the unspliced values, for example, the EM model posits a normal likelihood $p(u_{ng} | t_{ng}, k_{ng})$ similar to veloVI but without the latent cell state z_n , and learns posterior distributions $q(t_{ng} | u_{ng}, s_{ng})$ and $q(k_{ng} | u_{ng}, s_{ng})$.

Under the EM model, the posterior distributions are Dirac delta distributions, and the corresponding posterior predictive is expressed as

$$p(u_{ng}^* | u_{ng}, s_{ng}) = \int p_{\theta}(u_{ng}^* | t_{ng}, k_{ng}) q_{\phi}(t_{ng}, k_{ng} | u_{ng}, s_{ng}) dk_{ng} dt_{ng}.$$

Jianhua Xing

Reviewer #2 (Remarks to the Author):

The authors introduce veloVI, a deep learning model to infer RNA velocity parameters from single cell RNA-seq data using an adaptation of the scVI VAE framework. In brief, the authors provide spliced and unspliced count matrices as input, infer a latent set of RNA velocity and cell state parameters, and reconstruct spliced and unspliced count matrices from the latent state along with point-estimates of RNA velocity for each cell and gene sampled from the latent posterior. In comparisons to previous RNA velocity models, the authors report that veloVI yields velocity estimates that more accurately reflect simulations, are more robust to pre-processing choices, and enable estimates of RNA velocity uncertainty that aid interpretation.

The manuscript is well-written, the model is well reasoned, and the method represents a step forward for the RNA velocity inference field by introducing one of the first methods for quantifying velocity uncertainty in a robust manner. As presented, the model is compelling, but additional experiments are necessary to validate the full claims of the manuscript.

We thank the reviewer for the positive evaluation and the suggested interesting new experiments. Below we provide our point-by-point response to the reviewer's comments.

Major comments:

2.1 Comparison of veloVI and EM accuracy

In Fig. 1D the velocity inference results from veloVI and the EM algorithm are not particularly well correlated ($r^2 \approx 0.4$ by eye). Both procedures can't be correct in this case. Simulation results suggest that veloVI is marginally better than the EM algorithm, but **both** algorithms are highly correlated with simulated ground truth in that context (Fig. S2B). This suggests the simulations are not representative of any of the real datasets, and it's hard to know from these results alone if veloVI is actually more or less accurate than the EM approach.

The authors should include (1) experiments with more realistic simulations that reflect the disparity between veloVI and EM seen in all real datasets shown here, or ideally, (2) experiments in real data that have metabolic labels, allowing for an orthogonal, molecular method of inferring velocity that may better represent real biology. See scSLAM-seq (Erhard 2019), sci-Fate (Cao 2020), NASC-seq (Hendriks 2019), scNT-seq (Qiu 2020) as example datasets.

We thank the reviewer for this comment. While in Figure 1e, we demonstrated some cases (*Sulf2*, *Top2a*) in which the EM model's velocity estimates are less accurate than veloVI's (explaining in part the moderate correlation overall in Fig 1d), we agree that the current presentation does not comprehensively show the accuracy gains with veloVI. To address this comment we have made two major additions to the benchmarking:

1. More realistic simulations
2. Orthogonal validation with FUCCI cell cycle data

More realistic simulations

Here, we fit both veloVI and the *EM model* to the Pancreas dataset and recorded the estimated rate parameters. We then simulated data with these rate parameters (using the same framework as our original simulation, which was first described in the scvelo manuscript). We assessed the ability of veloVI and the EM model to recover the ground truth velocity and time and found that veloVI was significantly more accurate than the EM model, even when considering rates derived from the EM model. We attribute this result to the suboptimal time assignment of the EM model, which uses a grid of candidate time values, in addition to its overall heuristic inference procedure. In contrast, veloVI has an improved gradient-based optimization routine. These results were added to Supplementary Figure 2 as well as in the manuscript text:

Similarly, we validated that veloVI's inferred latent time and velocity correlate significantly better with ground truth compared to EM estimates when simulating data with parameters previously estimated on real data (Methods, Supplementary Fig. 2b).

Orthogonal validation with FUCCI cell cycle data

For an orthogonal validation, we applied both veloVI and the EM model to two cell cycle datasets (RPE-1-FUCCI cells and U2OS-FUCCI cells) with two opportunities for orthogonal validation:

1. For both datasets, the FUCCI system tracks the cell cycle progression through protein levels of key cell cycle regulators. The constructed cell cycle score is based on protein levels.
2. For one of the datasets (RPE-1-FUCCI cells), labeling with 5-ethynyl-uridine (EU) labeled transcripts

With respect to the first source of "ground truth" (FUCCI), we show that at the gene-level, veloVI is more accurate in estimating the sign of the velocity. These results demonstrate that veloVI's velocity directions are more consistent with the known sequence of states in the cell cycle. Additionally, we

showed that veloVI's estimated velocity is smoother (i.e., less noisy) with respect to the ground truth ordering of the cells by fitting Generalized Additive Models (GAMs). These results suggest that veloVI's velocities are more suitable for studying gene-level dynamics. We added the respective results as Supplementary Figure 9 as well as in the manuscript text:

To further validate the accuracy of veloVI, we compared veloVI and the *EM model* on cell cycle datasets of RPE1- and U2OS-FUCCI cells (Battich et al. 2020; Mahdessian et al. 2021) as it offers orthogonal validation of directionality/time via a protein-derived cell cycle score (Supplementary Fig. 9a). Compared to the *EM model*, veloVI achieves a higher velocity consistency (Supplementary Fig. 9b). We also tested whether the direction of the velocity at the gene-level aligns with a ground truth heuristic based on the cell cycle (Methods). As before, veloVI yielded consistent results and outperformed the *EM model* (RPE1 (resp. U2OS): 55% (resp. 68%) genes have higher velocity sign accuracy under veloVI; Supplementary Fig. 9c). As a complementary validation of these findings, we confirmed that the velocities of individual genes inferred by veloVI change more smoothly (i.e., are less noisy) with respect to the ground truth "time" compared to the *EM model* (RPE1 (resp. U2OS): 74% (resp. 65%) genes have higher R^2 under veloVI; Supplementary Fig. 9d-f; Methods).

As for directly leveraging the metabolic labeling information for an enhanced "ground truth" velocity estimate (either via this dataset, or one of the other technologies mentioned by the reviewer), we emphasize that this is non-trivial as demonstrated by the models/inference procedures proposed in *Dynamo* (Qiu et al. 2022). This is due to requiring various new assumptions about the dynamical process of total RNA and newly labeled RNA, which in turn requires a different estimation of rate parameters that may not be directly comparable to the rate parameters underlying veloVI.

In principle, we can compare the output of veloVI to the output of *Dynamo's* velocity routines, however, there are also important differences upstream of estimation related to data preprocessing that further challenge any direct comparison. In Supplementary Note 4, we discuss extensions of veloVI to infer rates from metabolically labeled transcripts as future work and refer to this in the discussion of the main text.

2.2 Velocity consistency comparisons

Fig. 1D (center panel) shows that veloVI has a higher "velocity consistency" across all datasets. Is this necessarily a desirable feature? In the simplest case, a kNN smoothing post-processing step on inferred

velocities would increase the consistency for any inference procedure. This only seems beneficial if it's abundantly clear that veloVI's estimates are *also* more likely to be accurate (See Major Point 1). The authors should comment on why higher consistency is always desirable and ideally demonstrate a case where known biology is better reflected using veloVI's more consistent estimates than the prior EM algorithm's estimates.

We agree with the reviewer that consistency indeed shows only one side of the coin, in a manner analogous to specificity (vs. sensitivity). In that sense, it can therefore be viewed as a necessary but not as a sufficient criterion for a successful analysis. The idea of smoothing is interesting, however, it comes post-hoc. The key point here is that smoothed values are generated in a principled manner by veloVI, without needing a post-hoc procedure.

In response to the previous reviewer comment, we demonstrated that veloVI's latent time and velocity estimates are more accurate than their EM counterparts according to simulations and on a real cell cycle dataset. The cell cycle contains a clear, well-defined direction of change characterized by the cell cycle score. Consequently, cells of comparable cell cycle scores share a common directionality, *i.e.*, the velocity consistency is expected to be high. Comparing the results of veloVI and the *EM model* shows again a higher consistency for our newly proposed model (as discussed in point 2.1 and shown in Supplementary Figure 9).

To highlight and demonstrate another case where known biology is better reflected using veloVI's more consistent estimates, we also refer to Figure 1e. In this case, the shown cell types with low expression (repression steady-state) are expected to exhibit small velocities. Indeed, veloVI's estimates are small whereas the *EM model* yields large velocities. We attribute this fact to veloVI's increased consistency.

2.3 Validating velocity uncertainty estimates

Estimating the uncertainty in velocity vectors is a major contribution, as prior methods in the literature for intrinsic uncertainty are expensive. However, it's difficult to determine from the current results alone if the uncertainty estimates provided by veloVI are well-calibrated. Many deep learning models are known to suffer from poor calibration absent specific techniques to improve uncertainty estimates, so it's not clear from first principles that veloVI uncertainty estimates are correct. The authors should

include experiments comparing the estimates of uncertainty from veloVI to other metrics of uncertainty derived by e.g. simulating data with different uncertainty on velocity parameters, Monte Carlo sampling of reads within each cell as an empirical estimate of uncertainty, or injecting random noise into data to show that veloVI estimates reflect increased uncertainty.

We thank the reviewer for this comment. We have included experiments in which we perturb (in silico) the pancreas dataset at different strengths and measure the corresponding uncertainty metrics. We include three different types of perturbations:

1. Downsampling the per cell transcript libraries to various levels, capturing sampling noise via sequencing depth and capture efficiency of RNA
2. Subsampling for a subset of genes the unspliced counts (via binomial thinning), capturing bias in the detection of unspliced RNA
3. Random multiplicative noise

We applied these perturbations at various strengths and found that the intrinsic uncertainty increased with perturbation strength. We found the same relationship for the extrinsic uncertainty except in the case of perturbation (1), which required a higher strength to result in an increase in uncertainty. The results are presented in Supplementary Figure 11.

We added this to the results section as follows:

To further understand what aspects of the data these uncertainty metrics capture, we (in silico) perturbed the pancreas dataset by either: (1) downsampling the total counts of each cell to mimic changes in sequencing depth and capture efficiency, (2) subsampling unspliced counts for a subset of genes to mimic the biased capture of unspliced molecules, or (3) adding random multiplicative noise to each abundance value (Methods). We applied each perturbation at various strengths and found that for each perturbation source, the intrinsic uncertainty increased with the perturbation strength. We found a similar response for the extrinsic uncertainty except in the case of total count downsampling, which required a high strength to shift the extrinsic uncertainty (Supplementary Fig. 11). These results suggest that the uncertainty metrics can capture random noise in the data, as well as bias in how the transcripts are measured.

To summarize, these results demonstrate that uncertainty metrics can capture relevant sources of noise and bias in the data.

2.4 Permutation score

The permutation score is very clever, and the results in Fig. 2d are compelling evidence it captures some real measure of how "dynamical" a cell system is. However, it's unclear how users should apply this score. Is it a pre-filtering metric for individual input genes? Is it a way to exclude whole clusters from analysis? Whole datasets? The authors should provide a clearer description of how and when this score is useful for downstream users of veloVI, and include experiments to demonstrate that utility. As it stands, the score is more of a curiosity than a useful tool.

We thank the reviewer for the positive evaluation of the permutation score. We have added text that describes explicit usage of the permutation score in practice (see blue text below), which also refers to use cases that were included in Supplementary Notes 1 and 2. To summarize:

1. Permutation score densities similar to the given negative control cases suggest that the corresponding datasets may not be suitable for RNA velocity and should potentially be excluded from further analysis. The reference set of permutation score densities is available via the accompanying software for users to compare against (see https://velovi.readthedocs.io/en/latest/api/reference/velovi.get_permutation_scores.html#velovi.get_permutation_scores).
2. Excluding entire clusters is dataset dependent, but a valid use case of the permutation score. Transient cell types of a lineage can be identified and retained. However, outlier clusters can also be identified and excluded.
 - a. We showcase and discuss this outlier scenario in the case of dendritic cells of the PBMC dataset where transition dynamics are not expected (Supplementary Figure 18f). Here, dendritic cells had genes with relatively high permutation scores, but transcriptionally similar cell types like monocytes had low scores for these genes. This does not support a monocyte-dendritic cell differentiation path in this dataset.
 - b. In the mouse developing dentate gyrus, cell types like OL, OPC, radial glial, and nIPC are not affected by permutation and can also be excluded (Supplementary Figure 19e).

Conversely, in the neuroblast-granule lineage (Supplementary Fig. 19f), we identified a substantial number of genes that were affected by permutation. This suggests that the presence of dynamics was identified by the model here.

3. The permutation procedure identifies genes consistent and inconsistent with transient dynamics. At the gene-level, the permutation score can be used for exploratory analysis, but we caution using it explicitly to filter genes before applying techniques that summarize velocities at the cellular level (e.g., 2d visualization or CellRank (Lange et al. 2022)) as this can create a feedback loop that results in creating a desired outcome. Generally, we propose using the score holistically, as we demonstrate in the point above, and to use this score in a transparent way when reporting results (i.e., reporting genes that do and do not support final results).

Overall, we believe this technique can be very helpful in explaining why (or why not) RNA velocity is working for a given dataset.

We added the following section to the manuscript:

In the accompanying code to this manuscript, we provide these permutation score densities as a resource for users of RNA velocity, which will enable the datasets we analyzed here to serve as references for the score distribution, and thus as a systematic approach to measure the overall transient dynamics of a dataset. For example, datasets exhibiting similar permutation score distributions as the given negative control cases (e.g., via kurtosis or skew) are not suitable for RNA velocity analysis with current models.

In Supplementary Notes 1 and 2, we provide case studies outlining how veloVI can be used in practice on peripheral blood mononuclear cells (PBMCs; negative control) and mouse developing dentate gyrus (partial positive control). These demonstrations synthesize veloVI's uncertainty quantification and permutation procedure along with the velocity coherence. When applying the permutation procedure, we were able to provide further evidence for the lack of transient populations in the case of PBMCs (Supplementary Note 1), as well as identify transient populations of neuroblasts and granule immature cells for many genes in dentate gyrus (Supplementary Note 2). Taken together, these results demonstrate that the permutation score is also useful for identifying cell populations that lack detectable transient dynamics.

Minor comments:

2.5 Fig. 1A is beautiful, but difficult to parse with the current legend. Readers would benefit from definitions of π and k_{ng} in the legend or inset alongside the panels.

We updated Figure 1A accordingly to improve the clarity of the presentation.

2.6 The authors should consider including the kNN smoothing + scaling preprocessing steps in Fig. 1A. This is impactful for interpreting the results.

We updated Figure 1A such that it now explicitly includes the preprocessing step.

2.7 The explanation of the permutation score in Fig. 2c is highly unclear. Fig. 2c was required for me to understand the procedure, but readers with less RNA velocity experience might not grok it from the plot. I'd suggest rephrasing to explain the intuition in the main text, and adding more clear explanation in the main methods for which values are being permuted. Maybe provide a sample set of vectors running through the procedure in Supplemental Methods.

We agree that the explanation could be improved. Therefore, we have rewritten the description in the results section:

We reasoned that the model fit of genes showing only steady-state dynamics would be robust to a permutation of the data while the model fit of genes with transient populations would worsen. Specifically for every gene, cell type, and species (spliced/unspliced) independently, we permuted the abundances of cells in a manner equivalent to shuffling cell barcodes. Subsequently, we passed this perturbed dataset through the veloVI model's trained encoder and decoder and recorded the absolute error of the fit grouped by genes and cell types. We then used the t-test statistic to compare the mean absolute error in each cell-type-gene group between the perturbed and original dataset (Supplementary Fig. 13, Methods).

We have also included a supplementary figure (Supplementary Figure 13) with a visual depiction of how the score is computed.

References

- Battich, Nico, Joep Beumer, Buys de Barbanson, Lenno Krenning, Chloé S. Baron, Marvin E. Tanenbaum, Hans Clevers, and Alexander van Oudenaarden. 2020. "Sequencing Metabolically Labeled Transcripts in Single Cells Reveals mRNA Turnover Strategies." *Science* 367 (6482): 1151–56.
- Lange, Marius, Volker Bergen, Michal Klein, Manu Setty, Bernhard Reuter, Mostafa Bakhti, Heiko Lickert, et al. 2022. "CellRank for Directed Single-Cell Fate Mapping." *Nature Methods* 19 (2): 159–70.
- Mahdessian, Diana, Anthony J. Cesnik, Christian Gnann, Frida Danielsson, Lovisa Stenström, Muhammad Arif, Cheng Zhang, et al. 2021. "Spatiotemporal Dissection of the Cell Cycle with Single-Cell Proteogenomics." *Nature* 590 (7847): 649–54.
- Qiu, Xiaojie, Yan Zhang, Jorge D. Martin-Rufino, Chen Weng, Shayan Hosseinzadeh, Dian Yang, Angela N. Pogson, et al. 2022. "Mapping Transcriptomic Vector Fields of Single Cells." *Cell* 185 (4): 690–711.e45.

Decision Letter, first revision:

Our ref: NMETH-BC50125A

6th Apr 2023

Dear Dr. Yosef,

Thank you for submitting your revised manuscript "Deep generative modeling of transcriptional dynamics for RNA velocity analysis in single cells" (NMETH-BC50125A). It has now been seen by the original referees and their comments are below. The reviewers find that the paper has improved in revision, and therefore we'll be happy in principle to publish it in *Nature Methods*, pending minor revisions to satisfy the referees' final requests and to comply with our editorial and formatting guidelines.

TRANSPARENT PEER REVIEW

Nature Methods offers a transparent peer review option for new original research manuscripts submitted from 17th February 2021. We encourage increased transparency in peer review by

publishing the reviewer comments, author rebuttal letters and editorial decision letters if the authors agree. Such peer review material is made available as a supplementary peer review file. **Please state in the cover letter 'I wish to participate in transparent peer review' if you want to opt in, or 'I do not wish to participate in transparent peer review' if you don't.** Failure to state your preference will result in delays in accepting your manuscript for publication.

ORCID

Sincerely,

Lin Tang, PhD
Senior Editor
Nature Methods

Reviewer #1 (Remarks to the Author):

The authors have addressed all my concerns.

Note on vector field reconstruction: The most important conceptual contribution of Qiu et al. is that with the estimated discrete RNA velocities (no matter from what methods, with or without metabolic labeling), one can learn the continuous vector field containing quantitative gene regulation information by formulating it as a machine learning problem. How to learn the vector field is only a technical question. There can be many algorithms to learn the vector field, and the Reproducible Kernel Hilbert Space method presented by Qiu et al. is only one of them.

Therefore, as pointed out by the authors in Note 4, with their new RNA velocities, there is no additional step needed to learn the vector field.

Reviewer #2 (Remarks to the Author):

The authors have performed a thorough response to reviewers. There are still remaining claims that would benefit from moderation and qualification in the main text, and these are outline point-by-point below. Overall, I am supportive of publishing the manuscript.

> More realistic simulations

- Additional simulations using real data to help set the parameters are appreciated, but don't fully address the original point. These simulations seem to likewise yield high correlation between **both** models and the ground truth, despite the realistic data being used to inform simulation parameters. They therefore can't both be largely correct, and only correlate with one another $r=0.4$ in a more realistic setting. The authors should spell out this limitation of their simulation study to readers in the main text when they present the results.
- The authors should include a statistical test to demonstrate that the veloVI model is superior to the EM model on these simulations. As of now, there are no error bars on the first set of simulation results (SFig 2a) or statistical tests on the new simulations (Sfig 2b), so it is challenging to assess the magnitude of improvements.

> Orthogonal validation with FUCCI cell cycle data

- At present, the Methods are a bit too vague on the sign-comparison procedure was performed. e.g. "We estimated this heuristic by first averaging the gene expression of all cells for a given cell cycle position" does not describe how the bin sizes were set. The results couldn't be reproduced from the methods description, and these analyses do not appear to be in the reproducibility repository. The authors should expand these methods appropriate to their importance in the overall evaluation of the method. These analyses are the best evidence in the paper to argue that veloVI is superior to the EM method in terms of accuracy.
- The authors should present the model estimates relative to the true values in the supplemental figure. For example, the sign agreements per gene should be displayed for both the VI and EM methods. At present, only the relative agreement of VI/EM methods is presented.
- Statistical tests should be performed. The VI vs. EM improvements appear modest in SFig. 9b, so it is important to confirm that these results are statistically robust. The authors may consider both binomial tests for per gene superiority as well as magnitude based tests that account for the degree of superiority/inferiority per gene.

> Velocity consistency comparisons

- The key point originally raised is that cells in a population may have similar current states, but truly heterogeneous velocities. It is therefore unclear that more consistent values are always better, regardless of the procedure used to generate them -- whether it be complex variational inference, or simple neighborhood averaging.
- It's unclear to me that the accuracy result in Fig. 1e can be directly attributed to "higher consistency" being a desired property of velocity estimates.
- As currently written, the manuscript argues that higher consistency is a generally desirable property, but this claim is not directly motivated by evidence.
- The authors should note in the main text that "higher consistency" is a heuristic evaluation procedure that relies on an assumption -- cells with more similar current states should have more

similar velocities in the datasets tested. The use of consistency would not be valid in cases where this assumption is broken.

Author Rebuttal, first revision:

Point-by-point response to the reviewers' comments

Deep generative modeling of transcriptional dynamics for RNA velocity analysis in single cells

Adam Gayoso^{1,*}, Philipp Weiler^{2,3,*}, Mohammad Lotfollahi², Dominik Klein², Justin Hong¹, Aaron Streets^{1,4,5}, Fabian J. Theis^{2,3,6,#}, Nir Yosef^{1,7,#}

* equal contribution

correspondence to fabian.theis@helmholtz-muenchen.de, niryosef@berkeley.edu

¹Center for Computational Biology, University of California, Berkeley, Berkeley, CA, USA

²Institute of Computational Biology, Helmholtz Center Munich, Munich, Germany

³Department of Mathematics, Technical University of Munich, Munich, Germany

⁴Department of Bioengineering, University of California, Berkeley, Berkeley, CA, USA

⁵Chan Zuckerberg Biohub, San Francisco, CA, USA

⁶TUM School of Life Sciences Weihenstephan, Technical University of Munich, Munich, Germany

⁷Department of Electrical Engineering and Computer Sciences, University of California, Berkeley, Berkeley, CA, USA

In the following, we present our response to the reviewers comments. We give **comments (black)**, **point-by-point answers (green)** to the questions and in parts **copy parts of the text or specific panels (blue)**, which directly correspond to comments or reference to them.

Editor comments:

Thank you for submitting your revised manuscript "Deep generative modeling of transcriptional dynamics for RNA velocity analysis in single cells" (N METH-BC50125A). It has now been seen by the original referees and their comments are below. The reviewers find that the paper has improved in revision, and therefore we'll be happy in principle to publish it in Nature Methods, pending minor revisions to satisfy the referees' final requests and to comply with our editorial and formatting guidelines.

TRANSPARENT PEER REVIEW

Nature Methods offers a transparent peer review option for new original research manuscripts submitted from 17th February 2021. We encourage increased transparency in peer review by publishing the reviewer comments, author rebuttal letters and editorial decision letters if the authors agree. Such peer review material is made available as a supplementary peer review file. Please state in the cover letter 'I wish to participate in transparent peer review' if you want to opt in, or 'I do not wish to participate in transparent peer review' if you don't. Failure to state your preference will result in delays in accepting your manuscript for publication.

Please note: we allow redactions to authors' rebuttal and reviewer comments in the interest of confidentiality. If you are concerned about the release of confidential data, please let us know specifically what information you would like to have removed. Please note that we cannot incorporate redactions for any other reasons. Reviewer names will be published in the peer review files if the reviewer signed the comments to authors, or if reviewers explicitly agree to release their name. For more information, please refer to our FAQ page.

We thank the reviewers for their comments and further recommendations and feedback. Based on the reviewer's feedback and the shared checklist, we have made the following main revisions:

- 1. Reformatting the manuscript.** We restructured the figures to accommodate the manuscript being published as an Article. The original Figure 1 was split into two; previous panels a and b (now Figure 1), and panels c-d (now Figure 3). Formerly Supplementary Figure 2b and Supplementary Figure 9a-c, f have been reorganized into Figure 2. As requested by Reviewer #2, Supplementary Figure 2b and Supplementary Figure 9c have been updated to include statistical test for significance. Figure 2 is now Figure 4. The results of Supplementary Figure 15a-e are now presented as Figure 5. Supplementary Figures 1, 3, 10, 11, 12, 13, 14 are now Extended Data Figures. All remaining panels and Supplementary Figures have been updated or kept as Supplementary Figures. We also improved clarity in the methods with mathematical notation and in figure legends.
- 2. Clarity of simulations and velocity consistency results.** We have added statistical tests for simulation and cell cycle results, which show a statistically significant improvement of veloVI over the EM model. We have also included writing in the main text that contextualizes these results.

We wish to participate in transparent peer review, and as requested, we provide twitter handles for making announcements (@adamgayoso, @PhilippWeiler7, @fabian_theis, @YosefLab).

Please find our point-by-point response to the reviewers' comments below, with references to the revisions in the main text (new additions in blue), figures and supplemental material.

Reviewer #1 (Remarks to the Author):

The authors have addressed all my concerns.

Note on vector field reconstruction: The most important conceptual contribution of Qiu et al. is that with the estimated discrete RNA velocities (no matter from what methods, with or without metabolic labeling), one can learn the continuous vector field containing quantitative gene regulation information by formulating it as a machine learning problem. How to learn the vector field is only a technical question. There can be many algorithms to learn the vector field, and the Reproducible Kernel Hilbert Space method presented by Qiu et al. is only one of them.

Therefore, as pointed out by the authors in Note 4, with their new RNA velocities, there is no additional step needed to learn the vector field.

We thank the reviewer for their comments. We made it more clear that this is only in comparison the the method proposed in Qiu et al. in the supplementary note as follows:

While veloVI is trained on a given dataset, unobserved data can be passed to the generative model as well. Consequently, compared to the vector field inference approach proposed in Dynamo [28], no additional step is required to infer vector fields for unobserved data.

Reviewer #2 (Remarks to the Author):

The authors have performed a thorough response to reviewers. There are still remaining claims that would benefit from moderation and qualification in the main text, and these are outline point-by-point below. Overall, I am supportive of publishing the manuscript.

> More realistic simulations

- Additional simulations using real data to help set the parameters are appreciated, but don't fully address the original point. These simulations seem to likewise yield high correlation between **both** models and the ground truth, despite the realistic data being used to inform simulation parameters. They therefore can't both be largely correct, and only correlate with one another $r=0.4$ in a more realistic setting. The authors should spell out this limitation of their simulation study to readers in the

main text when they present the results.

We agree with the reviewer that these simulations reflect an idealized scenario. We also note that the simulation more closely follows the *EM model's* generative process as genes are simulated independently (no low rank time factorization as veloVI assumes), therefore, we see this as an important demonstration of the advantage of veloVI over the *EM model*, even with the observation that performance deviates from real-world data. We have included the following text in the main text:

It is important to note that these simulations reflect an idealized scenario as cells are simulated via the EM model generative process, which assumes gene-wise independence, induction followed by repression states, and a single lineage (Methods). Nonetheless, veloVI outperforms the *EM model* even in these *EM*-favorable conditions.

- The authors should include a statistical test to demonstrate that the veloVI model is superior to the EM model on these simulations. As of now, there are no error bars on the first set of simulation results (Sfig 2a) or statistical tests on the new simulations (Sfig 2b), so it is challenging to assess the magnitude of improvements.

We have added boxplots to Supplementary Figure 2 and performed statistical tests, which are denoted on the boxplot. We found the difference to be significant under a two-sided Welch's t-test (p-value < 0.001).

> Orthogonal validation with FUCCI cell cycle data

- At present, the Methods are a bit too vague on the sign-comparison procedure was performed. e.g. "We estimated this heuristic by first averaging the gene expression of all cells for a given cell cycle position" does not describe how the bin sizes were set. The results couldn't be reproduced from the methods description, and these analyses do not appear to be in the reproducibility repository. The authors should expand these methods appropriate to their importance in the overall evaluation of the method. These analyses are the best evidence in the paper to argue that veloVI is superior to the EM method in terms of accuracy.

We extended the description of how the signs of velocities were identified, and compared. Specifically, we report the average number of cells per cell cycle position, and emphasize that these are defined by the original studies.

If a "ground truth" cellular ordering, for example, a cell cycle score as in refs. [13, 30], is given, we can make use of this source of information to estimate "ground truth" velocities \hat{v} via finite differences. We estimated this heuristic by first taking the median per gene of the first-order moment smoothed spliced RNA abundance of all cells at a given cell cycle position p_i which we denote by $\bar{s}^{(i)}$. Then, assuming the p_i are ordered ($p_i < p_{i+1}$), $\hat{v}^{(i)}$ is defined as

$\begin{equation}$

$$\hat{v}^{(i)} \propto \bar{s}^{(i+1)} - \bar{s}^{(i)}$$

$\end{equation}$

Finally, we compared the sign of all ground truth velocities with their inferred counterparts of veloVI and the EM model (which are aggregated per position in the same way) by computing the sign accuracy per gene. The sign accuracy, which is the fraction of times the signs agree, accounts for positive velocity, negative velocity, and zero velocity. As a baseline, we included a random predictor that chose positive, negative, or zero velocity with equal probability. The scEU-seq cell cycle data (RPE1-FUCCI cells)~\cite{battich2020sceu} included, on average, 9.63 (standard deviation 7.01) observations per cell cycle position and the U2OS-FUCCI~\cite{mahdessian2021spatiotemporal} dataset provided 1.15 (standard deviation 0.36) observations per cell cycle position. In the case of the U2OS-FUCCI dataset, the ground truth ordering was derived by the original authors using a polar regression on the scatter plot of the two FUCCI protein markers. In the case of the RPE1-FUCCI cells, the ground truth ordering was derived by the original authors using a pseudotime method on the FUCCI protein marker values.

Furthermore, we have updated the reproducibility repository to include all of the latest results.

- The authors should present the model estimates relative to the true values in the supplemental figure. For example, the sign agreements per gene should be displayed for both the VI and EM methods. At present, only the relative agreement of VI/EM methods is presented.

We replaced the relative agreement by the model-specific sign-comparison with ground truth. Additionally, we added the performance of a random predictor, and tested for significant difference between the performance of the methods.

- Statistical tests should be performed. The VI vs. EM improvements appear modest in SFig. 9b, so it is important to confirm that these results are statistically robust. The authors may consider both binomial tests for per gene superiority as well as magnitude based tests that account for the degree of superiority/inferiority per gene.

We tested for statistical significance of the difference in performance when inferring rates of the real-data-informed simulation, as well as the cell cycle study. We added these results in the figures and text as appropriate.

> Velocity consistency comparisons

- The key point originally raised is that cells in a population may have similar current states, but truly heterogeneous velocities. It is therefore unclear that more consistent values are always better, regardless of the procedure used to generate them -- whether it be complex variational inference, or simple neighborhood averaging.

- It's unclear to me that the accuracy result in Fig. 1e can be directly attributed to "higher consistency" being a desired property of velocity estimates.

- As currently written, the manuscript argues that higher consistency is a generally desirable property, but this claim is not directly motivated by evidence.

- The authors should note in the main text that "higher consistency" is a heuristic evaluation procedure that relies on an assumption -- cells with more similar current states should have more similar velocities

in the datasets tested. The use of consistency would not be valid in cases where this assumption is broken.

We thank the reviewer for the comments about velocity consistency. We have expanded on the assumptions made by velocity consistency in the main text:

This consistency measure quantifies the extent to which the velocities of cells with similar transcriptomic profiles (i.e., nearest neighbors) agree, and relies on the assumption that velocities change smoothly over the phenotypic manifold.

And the following additional text to the methods section:

This evaluation metric makes the assumption that better local consistency is inherently good, reflecting smooth changes in velocity over the phenotypic manifold. We note that this is a heuristic evaluation and the validity of this metric can be affected by, e.g., low density of similar cell states, misspecification of the k-nearest-neighbors graph due to only considering spliced RNA, etc.

Regarding Figure 1e, in the text we directly attribute this to veloVI estimating an expected velocity quantity, and do not explicitly attribute this to velocity consistency:

In the case of veloVI, Beta cells had nearly zero velocity, reflecting their belonging to the putative repression steady state for this gene. We attribute this result to veloVI's velocity directly marginalizing over the latent cell representations, which explicitly incorporates the probability that a cell belongs to induction, repression, or their respective steady states (Methods).

Final Decision Letter:

8th Aug 2023

Dear Professor Yosef,

I am pleased to inform you that your Article, "Deep generative modeling of transcriptional dynamics for RNA velocity analysis in single cells", has now been accepted for publication in Nature Methods. The paper will be published online before being published in a print issue. The received and accepted dates will be 12th Aug 2022 and 8th Aug 2023. This note is intended to let you know what to expect from us over the next month or so, and to let you know where to address any further questions.

Over the next few weeks, your paper will be copyedited to ensure that it conforms to Nature Methods style. Once your paper is typeset, you will receive an email with a link to choose the appropriate publishing options for your paper and our Author Services team will be in touch regarding any additional information that may be required.

Please note that *Nature Methods* is a Transformative Journal (TJ). Authors may publish their research with us through the traditional subscription access route or make their paper immediately open access through payment of an article-processing charge (APC). Authors will not be required to make a final decision about access to their article until it has been accepted. [Find out more about Transformative Journals](https://www.springernature.com/gp/open-research/transformative-journals)

Your paper will now be copyedited to ensure that it conforms to Nature Methods style. Once proofs are generated, they will be sent to you electronically and you will be asked to send a corrected version within 24 hours. It is extremely important that you let us know now whether you will be difficult to contact over the next month. If this is the case, we ask that you send us the contact information (email, phone and fax) of someone who will be able to check the proofs and deal with any last-minute problems.

Once your manuscript is typeset and you have completed the appropriate grant of rights, you will receive a link to your electronic proof via email with a request to make any corrections within 48 hours. If, when you receive your proof, you cannot meet this deadline, please inform us at rjsproduction@springernature.com immediately.

Once your paper has been scheduled for online publication, the Nature press office will be in touch to confirm the details.

Content is published online weekly on Mondays and Thursdays, and the embargo is set at 16:00 London time (GMT)/11:00 am US Eastern time (EST) on the day of publication. If you need to know the exact publication date or when the news embargo will be lifted, please contact our press office after you have submitted your proof corrections. Now is the time to inform your Public Relations or Press Office about your paper, as they might be interested in promoting its publication. This will allow them time to prepare an accurate and satisfactory press release. Include your manuscript tracking number NMETH-A50125B and the name of the journal, which they will need when they contact our office.

About one week before your paper is published online, we shall be distributing a press release to news organizations worldwide, which may include details of your work. We are happy for your institution or funding agency to prepare its own press release, but it must mention the embargo date and Nature Methods. Our Press Office will contact you closer to the time of publication, but if you or your Press Office have any inquiries in the meantime, please contact press@nature.com.

Nature Portfolio journals [encourage authors to share their step-by-step experimental protocols](https://www.nature.com/nature-research/editorial-policies/reporting-standards#protocols) on a protocol sharing platform of their choice. Nature Portfolio 's Protocol Exchange is a free-to-use and open resource for protocols; protocols deposited in Protocol Exchange are citable and can be linked from the published article. More details can found at www.nature.com/protocolexchange/about.

Please feel free to contact me if you have questions about any of these points. Thank you very much for publishing your paper at Nature Methods!

Best regards,

Lin Tang, PhD
Senior Editor
Nature Methods